# Coupling dynamic response of saturated soil with anisotropic thermal conductivity under fractional order thermoelastic theory

**Ying Guo[1,2,3], Chunbao Xiong[4], Wen Yu[5]\*, Jie Li[6], Jianjun Ma[1], Cui Du[1,7]**

**1** School of Civil Engineering, Henan University of Science and Technology, Luoyang, Henan, P.R. China, **2** School of Mechanical and Electrical Engineering, Henan University of Science and Technology, Luoyang, Henan, P.R. China, **3** State Key Laboratory of Hydraulic Engineering Simulation and Safety, Tianjin University, Tianjin, P.R. China, **4** School of Civil Engineering, Tianjin University, Tianjin, P.R. China, **5** School of Civil Engineering, Hebei University of Architecture, Zhangjiakou, Hebei, P.R. China, **6** No.4 Gas Production Plant, Changqing Oilfield Company, PetroChina, Xian, Shanxi, P.R. China, **7** School of Software Engineering, Chengdu University of Information Technology, Chengdu, Sichuan, P.R. China

\* yuwen810224@163.com

**Data Availability Statement:** All relevant data are within the paper and its Supporting Information files.

## Abstract

In this paper, a two-dimensional (2D) thermo-hydro-mechanical dynamic (THMD) coupling analysis in the presence of a half-space medium is studied using Ezzat's fractional order generalized theory of thermoelasticity. Using normal mode analysis (NMA), the influence of the anisotropy of the thermal conduction coefficient, fractional derivatives, and frequency on the THMD response of anisotropy, fully saturated, and poroelastic subgrade is then analyzed with a time-harmonic load including mechanical load and thermal source subjected to the surface. The general relationships among the dimensionless physical variables such as the vertical displacement, excess pore water pressure, vertical stress, and temperature distribution are graphically illustrated. The NMA method does not require the integration and inverse transformation, increases the decoupling speed, and eliminates the limitation of numerical inverse transformation. The obtained results can guide the geotechnical engineering construction according to different values of load frequency, fractional order coefficient, and anisotropy of thermal conduction coefficient. This improves the subgrade stability and enriches the theoretical studies on thermo-hydro-mechanical coupling.

## 1. Introduction

The interaction between deformation and temperature distribution can accurately describe the motion of the object while ignoring the pore water. The basic equation of the thermo-hydro-mechanical coupling process shows that the change of thermal state will be accompanied by the change of the displacement amount and excess pore water pressure, and vice versa.

Biot [1] proposed a unified description method for the linear coupled thermoelastic theory, which allows eliminating the shortcomings of the classical decoupling theory. It is known that the classical theories based on Fourier law have some drawbacks such as the infinite velocity of thermoelastic waves and poor description of the changes of the thermoelastic behavior at low

**Funding:** This work was supported in part by Natural Science Foundation of Henan Province (222300420153), Natural Science Foundation of Tianjin (S22YBJ1114), State Key Laboratory of Hydraulic Engineering Simulation and Safety, Tianjin University (HESS-2324), 2022 Heluo Young Talent Lifting Project (2022HLTJZC10), Scientific and Technological Project in Henan Province (212102310055), and Key Scientific Research Project of Henan Province (22A130004). The sponsor of Natural Science Foundation of Henan Province (222300420153), State Key Laboratory of Hydraulic Engineering Simulation and Safety, Tianjin University (HESS-2324), 2022 Heluo Young Talent Lifting Project (2022HLTJZC10), and Key Scientific Research Project of Henan Province (22A130004) is Ying Guo, who plays the roles of Conceptualization, Methodology, Supervision, Writing-original draft and Writing-review and editing in this paper. The sponsor of Natural Science Foundation of Tianjin (S22YBJ1114) is Chunbao Xiong, who plays the roles of Project administration and Supervision in this paper. Wen Yu without funding, she plays the role of Methodology and Validation in this paper. Jie Li without funding, she plays the role of Data curation in this paper. Jianjun Ma without funding, she plays the role of Project administration, and Writing-review and editing in this paper. The sponsor of Scientific and Technological Project in Henan Province (212102310055) is Cui Du, who plays the roles of Investigation and Visualization in this paper.

**Competing interests:** The authors have declared that no competing interests exist.

**Abbreviations:** Nomenclature $c_w$, Heat capacity of pore water; $c_s$, Heat capacity of solid grains; d, fdfffdfdf; E, Young's modulus in the subgrade; g, Gravity; $K_{11}, K_{33}$, Thermal conductivity; $k_d$, Permeability in the subgrade; m, Volumetric heat capacity; $n_0$, Porosity; p, Excess pore water pressure; $Q_0$, Thermal source; $q_0$, Mechanical load; T, Absolute temperature; $T_0$, Initial temperature; $u_i$, Displacement; Greek $\alpha$, Fractional order parameter; $0<\alpha\leq1$; $\alpha_s$, Thermal expansion coefficient of solid grains; $\alpha_u$, ; $\alpha_w$, Thermal expansion coefficient of pore water; $\beta_1$, $(3\lambda+2G)\alpha_s$; $\delta_{ij}$, Kronecker delta; $\varepsilon_{ij}$, Components of the strain tensor; $\theta$, $\theta = T−T_0$; $\lambda, G$, Lame's constants; $\mu_{ij}$, Poisson's ratio; $\rho$, Density; $\rho_w$, Density of pore water; $\rho_s$, Density of solid grains; $\sigma_{ij}$, Components of the stress tensor; $\tau$, Thermal relaxation time.

temperature and high flux. Therefore, several non-classical models, referred to as "generalized theories of thermoelastic", have been proposed. For instance, the generalized theory of thermoelasticity was proposed by Lord and Shulman [2] (the L-S generalized theory of thermoelasticity), which is based on the generalized Fourier law. It is also considered as a relaxation time for isotropic media. Green and Lindsay [3] proposed the G-L generalized theory of thermoelasticity which modifies the constitutive relations for stress-train and entropy by introducing two thermal relaxation time parameters. Green and Naghdi [4–6] developed three consistent logical theories: the viz. G-N I generalized theory of thermoelasticity, G-N II generalized theory of thermoelasticity, and G-N III generalized theory of thermoelasticity. The G-N-I theory assumes that the heat waves will travel at infinite velocities, while the Type II and III theories solve this problem by assuming that they will travel at finite velocities. However, no energy dissipation is observed in the G-N II theory. The G-N III model, which is mainly a combination of the first two model types, allows the energy dissipation. These are commonly used generalized theories of thermoelastic that can accurately describe the coupling effect among the thermal wave effect, temperature, and elastic fields. They can also represent the finite velocity heat wave in the medium, which is also known as the "second sound" of heat. Other theories, that can describe the coupling effect among the thermal wave effect, temperature, and elastic fields, also exist [7–9]. These theories have been widely used to solve many kinds of problems [10–17].

In practical engineering analysis and design, the thermo-mechanical response of poroelastic media cannot ignore the influence of liquid in the pore. Biot [18–20] proposed the governing equations of fluid-saturated porous elastic solids and the variational principle of thermoelastic coupling problems. In recent years, the thermo-mechanical properties of saturated porous media considering pore water have been widely studied. For instance, Booker and Savvidou [21, 22] developed analytical solutions to the basic problem of heat source buried deep in saturated soil, based on the Biot's thermoelastic theory. Bai [23, 24] studied lot of elastic and consolidation problems for fully fluid-saturated media that are caused by thermal impact, while considering the Biot's theory. Lu et al. [25] combined these generalized theories of thermoelastic and established the coupled governing equations with porous media for solving different isotropic saturated poroelastic problems [26, 27]. Guo et al. [28–30] studied two-dimensional thermo-hydro-elastic coupling problems for a series of isotropic, homogeneous, fully saturated, porous elastic and porous viscoelastic half-space subgrades. Shakeriaski et al. [31] studied the transient response of a porous medium affected by external traction using the G-L theory of thermoelasticity. Zhu et al. [32] studied the thermo-hydro-mechanical coupling of saturated porous deep-sea sediments under the action of mine cart vibration, based on the G-L generalized theory of thermoelasticity and Darcy's law.

The fractional differential equations have been widely used in many engineering fields. The concept of fractional calculus is a natural extension of classical mathematics, in which several definitions [33–35] have been proved to have certain self-similarity. This self-similarity allows it to be used in many application domains. Several definitions of fractional derivatives have been proposed and used in different fields.

With the rapid development of technology, fractional calculus has been introduced and evolved in the field of thermoelasticity. Caputo [36] derived the fractional order generalized theory of thermoelasticity by introducing the Riemann-Liouville fractional integral operator, which naturally introduces the initial conditions in the problem definition. Povstenko [37] proposed an uncoupled thermoelastic theory that can be used to describe the quasi-static state. Youssef et al. [38] proposed a fractional heat conduction equation with a fractional coefficient of $0<\alpha\leq2$ and established a new fractional generalized thermoelastic theory, based on the Riemann-Liouville fractional integral operator. Sherief et al. [39] modified the L-S generalized

theory of thermoelasticity and combined it with fractional calculus to develop a novel generalized thermoelasticity model. Youssef and Al-Lehaibi [40] developed a mathematical model which can be used to describe the thermal projectile of semiconductor solid spheres. They then solved a thermoporoelastic problem in the context of G-N theory with fractional-order parameters. Zhang and Ma [41] studied the nonlocal fractional order strain problem which is subject to moving heat source using the fractional order strain theory. Abouelregal et al. [42] proposed a generalized thermoelastic model with two-temperature characteristics: the heat transfer equation with Caputo-Fabrizio fractional differential operator and the phase lags. Abouelregal et al. [43] analyzed the thermoelastic vibrations of a nonlocal isotropic solid medium subjected to a pulsed heat flux based on the Caputo-Fabrizio fractional derivative generalized thermoelasticity. AL-Lehaibi [44] solved a two-dimensions thermoelastic problem for an isotropic and homogeneous nanobeam in terms of the thermoelasticity with one relaxation time and fractional-order strain theory. Sherief and Hussein [45] solved a two-dimensional thermoporoelastic problem for infinitely porous cylinders at certain boundary conditions, using the fractional order generalized thermo-poroelasticity theory. Hu et al. [46] used a fractional dual-phase-lag bio-thermoelastic model to solve the thermoelastic response of a biological tissue during hyperthermia treatment by a moving laser heating. Han et al. [47] studied the thermoelastic transient response of porous microplates subjected to thermal and stress shocks at the left boundary based on the Atangana-Baleanu fractional order generalized thermoelastic theory. This was performed by considering nonlocal effects and fractional order strain combined with the thermoelasticity theory of porous materials and the dual phase-lag heat conduction model. Dutta et al. [48] proposed a generalized thermo-diffusion process in a semi-infinite nonlocal fiber-reinforced double porous thermoelastic diffusive material with voids using the fractional-order Lord-Shulman thermo-elasto-diffusion theory.

The practice has shown that natural soil is formed by the long-term deposition process of rocks, which may lead to the structure anisotropy and the large difference of thermal physical properties in different directions, especially in the horizontal and vertical directions. Yue [49] studied a solution for the thermoelastic problem in vertically inhomogeneous media. Ai et al. [50] derived the 3D thermoelastic problem of layered media around a heat source and obtained its general solution. Ai and Wang [51] studied the three-dimensional thermodynamic behavior of layered materials with anisotropic thermal diffusivity in the Cartesian coordinate system. They then obtained the time-dependent thermal and mechanical responses of the material system. Ai and Wu [52] studied, using the Laplace-Hankel transforms, the impacts of the thermal diffusivity and permeability anisotropy on the thermal consolidation of multilayer porous thermoelastic media. Wang et al. [53] studied the time-dependent response of excess pore water pressure in subsurface saturated soils under the joint action of temperature load and mechanical load, based on the Laplace-Hankel transform and exact integral method. Das et al. [54] studies a three-dimensional thermoelastic problem for an anisotropic rectangular plate.

However, few studies tackle the thermo-hydro-mechanical coupling dynamic problem for anisotropic fully fluid-saturated and poroelastic semi-infinite media with mechanical load or thermal source. Therefore, this study focuses on the thermo-hydro-mechanical coupling dynamics of anisotropic saturated subgrade using the Ezzat's fractional generalized theory of thermoelasticity. Two-dimensional distributions of the on-dimensional excess pore water pressure, vertical displacement, vertical stress, and temperature are analytically derived using normal mode analysis, based on the Caputo fractional derivative of order $0 < \alpha \leq 1$. The impacts of the load frequency, fractional order parameter, and anisotropy of the thermal conduction coefficient on different physical variables in the subgrade are then detailed. The results obtained in this study have a wide application range in the field of geotechnical engineering.

## 2. Basic equations

In this study, a half-space subgrade with anisotropy of thermal conduction coefficient is considered, where the thermal source or mechanical load is subject to the surface. The constitutive equation is given by [55]:

$$\sigma_{ij,j} = \rho \ddot{u}_i \tag{1}$$

Note that in the sequel, a comma followed by a suffix denotes a derivative with respect to the material coordinate, a superposed dot denotes a derivative with respect to time, $i,j = x,z$ and $\sigma_{ij}$ are the components of the stress tensor, $\rho$ is the density of the medium, and $u_i$ denotes the displacement vector components.

The strain-displacement equation is given by:

$$\varepsilon_{ij} = \frac{1}{2}\left(u_{i,j} + u_{j,i}\right) \tag{2}$$

The constitutive equation of the THMD model can be expressed as:

$$\sigma_{ij} = 2G\varepsilon_{ij} + (\lambda e - \beta_1 \theta - p)\delta_{ij} \tag{3}$$

Where $p$ is the excess pore water pressure, $\theta$ is the increment of the temperature which is equal to $T–T_0$, $T_0$ is the initial reference temperature, and $T$ is the absolute temperature.

Based on Eqs (1)–(3), the constitutive equation of the THMD model can be expressed as:

$$Gu_{i,jj} + (\lambda + G)u_{j,ij} - \beta_1 \theta_{,i} - p_{,i} = \rho \ddot{u}_i \tag{4}$$

The heat conduction equation for a porous subgrade with fractional derivative, which is modified by the L-S generalized thermoelastic theory, can be expressed as:

$$K_{11}\frac{\partial^2 \theta}{\partial x^2} + K_{33}\frac{\partial^2 \theta}{\partial z^2} = \left(1 + \frac{\tau^\alpha}{\alpha!}\frac{\partial^\alpha}{\partial t^\alpha}\right)\left(m_i\dot{\theta} + \beta_{1x}T_0\dot{e}\right) \tag{5}$$

$$\frac{\partial^\alpha}{\partial t^\alpha}f(x,t) = \begin{cases} f(x,t) - f(x,0) & \alpha \to 0 \\ I^{1-\alpha}\dfrac{\partial f(x,t)}{\partial t} & 0 < \alpha < 1 \\ \dfrac{\partial f(x,t)}{\partial t} & \alpha = 1 \end{cases}$$

where $m = n_0\rho_w c_w + (1 - n_0)\rho_s c_s$ denotes the volumetric heat capacity in different directions, $n_0$ represents the porosity in different directions, $\rho_w$ and $\rho_s$ are respectively the solid grains and density of pore water, $c_w$ and $c_s$ are respectively the solid grains and heat capacity of pore water, $\tau$ denotes the thermal relaxation time, $K_{11}$ and $K_{33}$ are the heat conduction coefficients in the horizontal and vertical directions, respectively.

The mass conservation equation for anisotropic media is given by:

$$d(\alpha_w\dot{\theta} - \dot{e}) + \rho_w\ddot{e} + p_{,ii} = 0 \tag{6}$$

## 3. Problem formulation

The dynamic problem of anisotropy, fully fluid-saturation, and poroelasticity is studied using the Ezzat's fractional order generalized theory of thermoelasticity. A mechanical load (or a thermal source) is considered, and it is assumed that, as $z \to \infty$, the considered functions are

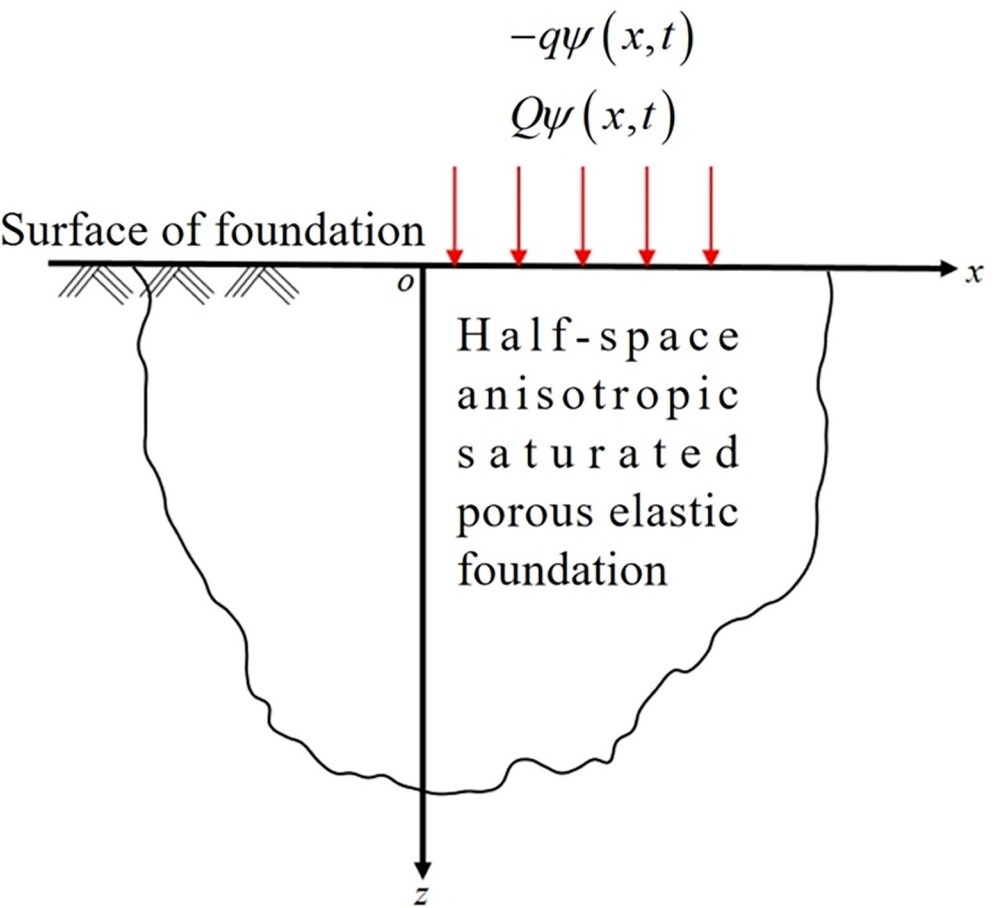

**Fig 1. Schematic illustration of the problem.**

bounded and vanished, as shown in Fig 1. In this study, the Cartesian coordinate system ($x,y,z$) and displacement component coordinate system $u_i = (u,0,w)$ are used.

The constitutive equation is given by:

$$
\begin{cases}
\sigma_x = (\lambda + 2G)e - 2G\dfrac{\partial w}{\partial z} - \beta_1\theta - p & (7a) \\[2mm]
\sigma_z = (\lambda + 2G)e - 2G\dfrac{\partial u}{\partial x} - \beta_1\theta - p & (7b) \\[2mm]
\sigma_{xz} = G\left(\dfrac{\partial u}{\partial z} + \dfrac{\partial w}{\partial x}\right) & (7c)
\end{cases}
\tag{7}
$$

The motion equation is given by:

$$
\begin{cases}
(\lambda + 2G)\dfrac{\partial^2 u}{\partial x^2} + (\lambda + G)\dfrac{\partial^2 w}{\partial x \partial z} + G\dfrac{\partial^2 u}{\partial z^2} - \beta_1\dfrac{\partial \theta}{\partial x} - \dfrac{\partial p}{\partial x} = \rho\dfrac{\partial^2 u}{\partial t^2} & (8a) \\[3mm]
(\lambda + 2G)\dfrac{\partial^2 w}{\partial z^2} + (\lambda + G)\dfrac{\partial^2 u}{\partial x \partial z} + G\dfrac{\partial^2 w}{\partial x^2} - \beta_1\dfrac{\partial \theta}{\partial z} - \dfrac{\partial p}{\partial z} = \rho\dfrac{\partial^2 w}{\partial t^2} & (8b)
\end{cases}
\tag{8}
$$

The following dimensionless variables are introduced to simplify the derivation process and make the derivation result universal [25, 56]:

$$(\tilde{x}, \tilde{z}, \tilde{u}, \tilde{w}) = V\eta_1(x, z, u, w) \quad (\tilde{t}, \tilde{\tau}) = V^2\eta_1(t, \tau)$$

$$\tilde{\theta} = \frac{\beta_1}{\lambda + 2G}\theta \quad \tilde{p} = \frac{1}{\lambda + 2G}p \quad \tilde{\sigma}_{ij} = \frac{1}{G}\sigma_{ij} \tag{9}$$

where $\eta_1 = \frac{m}{K_{11}}$ and $V = \sqrt{\frac{\lambda+2G}{\rho}}$.

According to Eq (9), Eqs (5)–(8) are rewritten in dimensionless forms (the asterisk is dropped for convenience) as follows:

$$\gamma^2 \frac{\partial^2 u}{\partial x^2} + (\gamma^2 - 1)\frac{\partial^2 w}{\partial x \partial z} + \frac{\partial^2 u}{\partial z^2} - \gamma^2 \frac{\partial \theta}{\partial x} - \gamma^2 \frac{\partial p}{\partial x} = \gamma^2 \frac{\partial^2 u}{\partial t^2} \tag{10}$$

$$\gamma^2 \frac{\partial^2 w}{\partial x^2} + (\gamma^2 - 1)\frac{\partial^2 u}{\partial x \partial z} + \frac{\partial^2 w}{\partial z^2} - \gamma^2 \frac{\partial \theta}{\partial z} - \gamma^2 \frac{\partial p}{\partial z} = \gamma^2 \frac{\partial^2 w}{\partial t^2} \tag{11}$$

$$\frac{\partial^2 \theta}{\partial x^2} + \iota \frac{\partial^2 \theta}{\partial z^2} = \left(1 + \frac{\tau^\alpha}{\alpha!}\frac{\partial^\alpha}{\partial t^\alpha}\right)\left(\frac{\partial \theta}{\partial t} + \iota_0\left(\frac{\partial^2 u}{\partial x \partial t} + \frac{\partial^2 w}{\partial z \partial t}\right)\right) \tag{12}$$

$$\frac{\partial^2 p}{\partial x^2} + \frac{\partial^2 p}{\partial z^2} + \iota_2 \frac{\partial \theta}{\partial t} - \iota_1\left(\frac{\partial^2 u}{\partial x \partial t} + \frac{\partial^2 w}{\partial z \partial t}\right) + \iota_3\left(\frac{\partial^3 u}{\partial x \partial^2 t} + \frac{\partial^3 w}{\partial z \partial^2 t}\right) = 0 \tag{13}$$

$$\sigma_x = \gamma^2 \frac{\partial u}{\partial x} + (\gamma^2 - 2)\frac{\partial w}{\partial z} - \gamma^2 \theta - \gamma^2 p \tag{14}$$

$$\sigma_z = (\gamma^2 - 2)\frac{\partial u}{\partial x} + \gamma^2 \frac{\partial w}{\partial z} - \gamma^2 \theta - \gamma^2 p \tag{15}$$

$$\tau_{xz} = \frac{\partial u}{\partial z} + \frac{\partial w}{\partial x} \tag{16}$$

where:

$$\iota = \frac{K_{33}}{K_{11}} \quad \iota_0 = \frac{T_0\beta_1^2}{m(\lambda+2G)} \quad \iota_1 = \frac{b}{\eta_1(\lambda+2G)} \quad \iota_2 = \frac{b\alpha_u}{\eta_1\beta_1} \quad \iota_3 = \frac{\rho_w}{\rho}\gamma^2 = \frac{\lambda+2G}{G}$$

## 4. Process of normal mode analysis

The normal mode analysis is an inexpensive technique which simulates the low and large amplitude motions. It solves the problem of discrete and truncation errors in the numerical inverse transformation and it can be divided into two parts without integral transformation and inverse transformation, which increases the decoupling speed. In addition, it can be used to characterize the macro-molecular flexibility to predict the directions of compositional changes and interpret the structural experimental data. This method is applied in various fields such as structural engineering, biological physics, molecular spectroscopy, and structural biology. In this study, the NMA weighted residual method is introduced. The solution of the considered variable can be decomposed into the following normal modes [57]:

$$(u, w, e, \sigma_{ij}, p)(x, z, t) = (u', w', e', \sigma'_{ij}, p')(z)\exp(\omega t + iax) \tag{17}$$

where frequency $\omega$ is a complex time constant, $i = \sqrt{-1}$ is the imaginary value, $a$ is the

number of waves along the $x$-direction, and $(u', w', e', \sigma'_{ij}, p')(z)$ represents the amplitudes of the field variables. Note that the value of $a$ is related to the wave number, wavelength, and frequency.

Substituting Eq (17) in Eqs (10)–(16) yields:

$$\left(\frac{d^2}{dz^2} - g_1\right)\theta'(z) = g_2 u'(z) + g_3 \frac{dw'(z)}{dz} \tag{18}$$

$$\left(\frac{d^2}{dz^2} - g_4\right)u'(z) + g_5 \frac{dw'(z)}{dz} - g_6\theta'(z) - g_6 p'(z) = 0 \tag{19}$$

$$\left(\frac{d^2}{dz^2} - g_7\right)w'(z) + g_8 \frac{du'(z)}{dz} - \frac{d\theta'(z)}{dz} - \frac{dp'(z)}{dz} = 0 \tag{20}$$

$$\left(\frac{d^2}{dz^2} - a^2\right)p'(z) - g_9 u'(z) - g_{10}\frac{dw'(z)}{dz} + g_{11}\theta'(z) = 0 \tag{21}$$

$$\sigma'_x(z) = ia\beta^2 u'(z) + (\beta^2 - 2)\frac{dw'(z)}{dz} - \beta^2\theta'(z) - \beta^2 p'(z) \tag{22}$$

$$\sigma_z(z) = (\beta^2 - 2)iau'(z) + \beta^2 \frac{dw'(z)}{dz} - \beta^2\theta'(z) - \beta^2 p'(z) \tag{23}$$

$$\tau'_{xz}(z) = \frac{du'(z)}{dz} + iaw'(z) \tag{24}$$

where:

$$g_1 = \frac{a^2 + \omega\left(1 + \frac{\tau^\alpha}{\alpha!}\omega^\alpha\right)}{\iota} \quad g_2 = \frac{ia\iota_0\omega\left(1 + \frac{\tau^\alpha}{\alpha!}\omega^\alpha\right)}{\iota} \quad g_3 = \frac{\iota_0\omega\left(1 + \frac{\tau^\alpha}{\alpha!}\omega^\alpha\right)}{\iota}$$

$$g_4 = \gamma^2 a^2 + \gamma^2\omega^2 \quad g_5 = (\gamma^2 - 1)ia \quad g_6 = \gamma^2 ia \quad g_7 = \frac{a^2 + \gamma^2\omega^2}{\gamma^2}$$

$$g_8 = \frac{(\gamma^2 - 1)ia}{\gamma^2} \quad g_9 = ia\iota_1\omega - ia\iota_3\omega^2 \quad g_{10} = \iota_1\omega - \iota_3\omega^2 \quad g_{11} = \iota_2\omega$$

Eqs (18)–(21) can be simplified to:

$$m_1\theta'(z) = g_2 u'(z) + m_2 w'(z) \tag{25}$$

$$m_3 u'(z) + m_4 w'(z) - g_6\theta'(z) - g_6 p'(z) = 0 \tag{26}$$

$$m_5 w'(z) + m_6 u'(z) - m_7\theta'(z) - m_7 p'(z) = 0 \tag{27}$$

$$m_8 p'(z) - g_9 u'(z) - m_9 w'(z) + g_{11}\theta(z) = 0 \tag{28}$$

The following eighth-order equation can be obtained by applying the elimination procedure on the system of Eqs (25)–(28):

$$(D^8 - B_1 D^6 + B_2 D^4 - B_3 D^2 + B_4)(u', w', \theta', p')(z) = 0 \tag{29}$$

Eq (29) can be factorized as:

$$(D^2 - k_i^2)^4 (u', w', \theta', p')(z) = 0 \tag{30}$$

where $k_i^2 (i = 1, 2, 3, 4)$ denotes the roots of Eq (30).

$$k^8 - B_1 k^6 + B_2 k^4 - B_3 k^2 + B_4 = 0 \tag{31}$$

The roots $\pm k_1$, $\pm k_2$, $\pm k_3$, and $\pm k_4$ in Eq (31) have the following relations:

$$k_1^2 + k_2^2 + k_3^2 + k_4^2 = B_1$$

$$k_1^2 k_2^2 + k_1^2 k_3^2 + k_1^2 k_4^2 + k_2^2 k_3^2 + k_2^2 k_4^2 + k_3^2 k_4^2 = B_2$$

$$k_1^2 k_2^2 k_3^2 + k_1^2 k_2^2 k_4^2 + k_1^2 k_3^2 k_4^2 + k_2^2 k_3^2 k_4^2 = B_3$$

$$k_1^2 k_2^2 k_3^2 k_4^2 = B_4$$

Using the normal mode analysis conditions ($u'(z) \to 0$, $w'(z) \to 0$, $\theta'(z) \to 0$, and $p'(z) \to 0$ as $z \to \infty$), the solution of Eqs (25)–(28) and (31) can be expressed as:

$$u'(z) = \sum_{n=1}^{4} M_{1n}(a, \omega) e^{-k_n z} \tag{32}$$

$$w'(z) = \sum_{n=1}^{4} {}'M_{1n}(a, \omega) e^{-k_n z} \tag{33}$$

$$\theta'(z) = \sum_{n=1}^{4} M_{1n}''(a, \omega) e^{-k_n z} \tag{34}$$

$$p'(z) = \sum_{n=1}^{4} M_{1n}'''(a, \omega) e^{-k_n z} \tag{35}$$

where $M_{1n}$, ${}'M_{1n}$, $M_{1n}''$, and $M_{1n}'''$ are parameters depending on $a$ and $\omega$.

By substituting Eqs (32)–(35) into Eqs (25)–(28), the relations of $M_{1n}$, ${}'M_{1n}$, $M_{1n}''$, and $M_{1n}'''$ become as follows:

$${}'M_{1n}(a, \omega) = H_{1n} M_{1n}(a, \omega) \tag{36}$$

$$M_{1n}''(a, \omega) = H_{2n} M_{1n}(a, \omega) \tag{37}$$

$$M_{1n}'''(a, \omega) = H_{3n} M_{1n}(a, \omega) \tag{38}$$

where:

$$H_{1n} = \frac{g_9 m_5 - m_4 m_7}{g_6 m_6 - m_3 m_7} \quad H_{2n} = \frac{g_2}{m_1} - \frac{m_2 H_{1n}}{m_1} \quad H_{3n} = \frac{g_9}{m_8} + \frac{m_9 H_{1n}}{m_8} - \frac{g_{11} H_{2n}}{m_8}$$

$$m_1 = D^2 - g_1 \quad m_2 = g_3 D \quad m_3 = D^2 - g_4 \quad m_4 = g_5 D \quad m_5 = D^2 - g_7$$

$$m_6 = g_8 D \quad m_7 = D \quad m_8 = D^2 - a^2 \quad m_9 = g_{10} D$$

The following can then be obtained:

$$w'(z) = \sum_{n=1}^{4} H_{1n} M_{1n}(a, \omega) e^{-k_n z} \tag{39}$$

$$\theta'(z) = \sum_{n=1}^{4} H_{2n} M_{1n}(a, \omega) e^{-k_n z} \tag{40}$$

$$p'(z) = \sum_{n=1}^{4} H_{3n} M_{1n}(a, \omega) e^{-k_n z} \tag{41}$$

Using the dimensionless parameters defined in Eq (17) and applying the normal mode analysis, the stress expressions defined in Eqs (22)–(24) can be written as:

$$\sigma'_{xx} = \sum_{n=1}^{4} H_{4n} M_{1n}(a, \omega) e^{-k_n z} \tag{42}$$

$$\sigma'_{zz} = \sum_{n=1}^{4} H_{5n} M_{1n}(a, \omega) e^{-k_n z} \tag{43}$$

$$\sigma'_{xz} = \sum_{n=1}^{4} H_{6n} M_{1n}(a, \omega) e^{-k_n z} \tag{44}$$

where:

$$H_{4n} = ia\beta^2 - k_n(\beta^2 - 2)H_{1n} - \beta^2 H_{2n} - \beta^2 H_{3n}$$

$$H_{5n} = (\beta^2 - 2)ia - k_n \beta^2 H_{1n} - \beta^2 H_{2n} - \beta^2 H_{3n}$$

$$H_{6n} = -k_n + iaH_{1n}$$

## 5. Boundary conditions of the problem

### 5.1 Mechanical load condition

1. (1) Stress condition:

$$\sigma_{xz} = 0 \quad \sigma_{zz} = -q_0 \psi(x, t) \tag{45}$$

where $q_0$ is the mechanical load

1. Thermal condition:

$$\theta = 0 \tag{46}$$

1. Excess pore water pressure condition:

$$p = 0 \tag{47}$$

## 5.2 Thermal source condition

1. (1) Stress condition:

$$\sigma_{xz} = 0 \quad \sigma_{zz} = 0 \tag{48}$$

1. Thermal condition:

$$\theta = Q_0 \psi(x, t) \tag{49}$$

where $Q_0$ is the thermal source and $\psi(x,t)$ is the distribution function of the mechanical load or thermal source on the *x-axis*, which can be written as:

$$\psi(x, t) = \psi'(a, \omega) \exp(\omega t + iax) \tag{50}$$

1. Excess pore water pressure condition:

$$p = 0 \tag{51}$$

## 6. Numerical results and discussion

A numerical example is used to demonstrate the high efficiency of the proposed analytical method. The Riemann-Liouville integral operator is taken into account in the Ezzat's fractional generalized thermoelastic model, and the dimensionless vertical displacement, excess pore water pressure, vertical stress, and temperature changes are then calculated. To study the THMD coupling of fluid-saturated porous roadbed, most of the used parameters are taken from [23, 58]. They are summarized as:

$$E = 6.0 \times 10^5 \text{N}/m^2 \quad \mu = 0.3 \quad G = E/[2(1 + \mu)] \quad k_d = 1.0 \times 10^{-8} \text{m/s}$$
$$\tau = 0.02 T_0 = 300\text{K} \quad \lambda = E\mu/[(1 + \mu)(1 - 2\mu)] \quad \alpha_s = 3.0 \times 10^{-5} K^{-1}$$
$$\rho_w = 1.0 \times 10^3 \text{kgm}^{-3} \quad n_0 = 0.4 \quad \alpha_w = 3.0 \times 10^{-4} K^{-1} \quad \omega = \omega_0 + i\zeta$$
$$e^{\omega t} = e^{\omega_0 t}(\cos\zeta t + i\sin\zeta t) \quad K = 0.582 \text{Wm}^{-1} K^{-1} s^{-1} \quad c_s = 973 \text{J} kg^{-1} K^{-1}$$
$$c_w = 4186 \text{J} kg^{-1} K^{-1} \quad \rho_s = 2.6 \times 10^3 \text{kgm}^{-3}$$

The other constants of the porous problem are given by:

$$a = 1.2 \quad \psi^* = 1$$

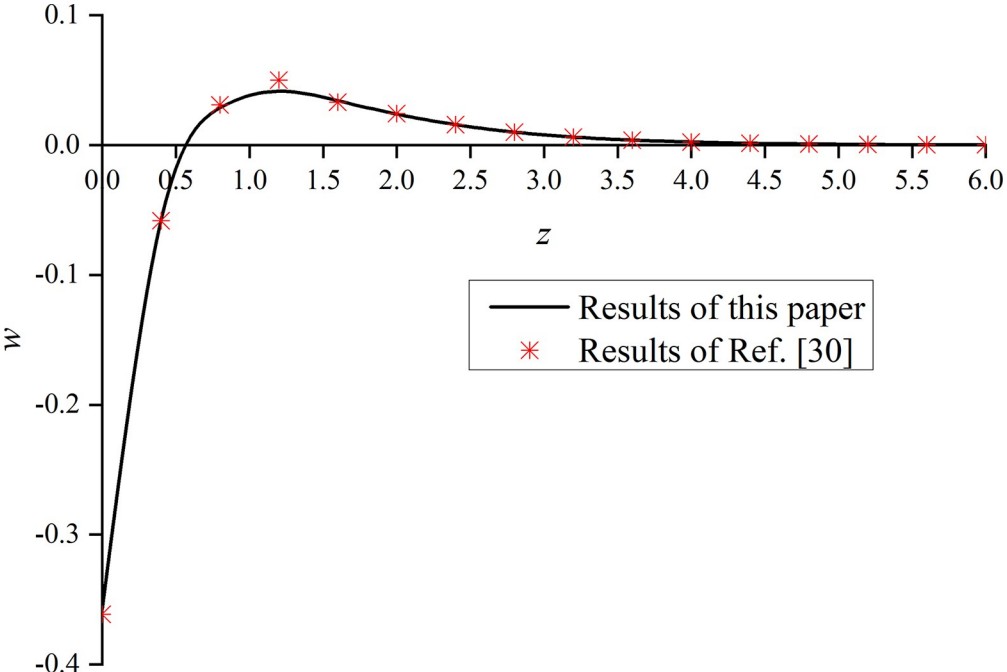

**Fig 2. Comparison between the non-dimension displacement and that presented in [30].**

In this paper, $\varphi = 1.0$ and $\alpha$ are set to 1. That is, the model is degraded to the dynamic response problem of isotropic media under the integer order generalized thermoelastic theory. In addition, the problem is solved using the boundary conditions presented in [30], which results are compared with the obtained dimensionless displacement, excess pore water pressure, vertical stress, and temperature under thermal shock. It can be seen from Figs 2–4 that, for $\varphi = 1.0$ and $\alpha = 1.0$, the results of the two studies are mainly consistent, and the different numerical calculations cause small errors, with maximum and average deviations of 5.07% and 2.5%, respectively. The correctness and rationality of the calculated results are further verified.

## 6.1 Effect of the anisotropy of the thermal conduction coefficient and fractional order parameter with mechanic load or thermal source

This section studies the impact of the anisotropy of heat conductivity and fractional order parameters on the physical variables under mechanical load and thermal sources. Figs 2–5 show the variation rules of dimensionless physical variables including the vertical displacement $w' = w/q$, excess pore water pressure $p' = p/q$, vertical stress $\sigma' = \sigma/q$, and temperature $\theta' = \theta/q$ in the subgrade, when its surface is affected by mechanic load. Figs 6–9 show the variations of the dimensionless vertical displacement $w'' = w/Q_0$, dimensionless excess pore water pressure $p'' = p/Q_0$, dimensionless vertical stress $\sigma'' = \sigma/Q_0$, and dimensionless temperature $\theta'' = \theta/Q_0$, in the case of a thermal source on the surface. Three different values of the thermal conduction coefficient anisotropy ($\varphi = 0.8$, $\varphi = 0.9$, and $\varphi = 1.0$) and two different values of the fractional order parameter ($\alpha = 0.1$ and $\alpha = 1.0$), are specified. These computations are performed for $x = 1.0$ at time $t = 0.5$.

Figs 6–9 show the influence of the variation of $\varphi$ and $\alpha$ on different dimensionless physical variables with mechanical load. The change of $\varphi$ has a certain influence on all the variables, except the dimensionless excess pore water pressure. In all the cases, $\varphi$ represents the ratio of

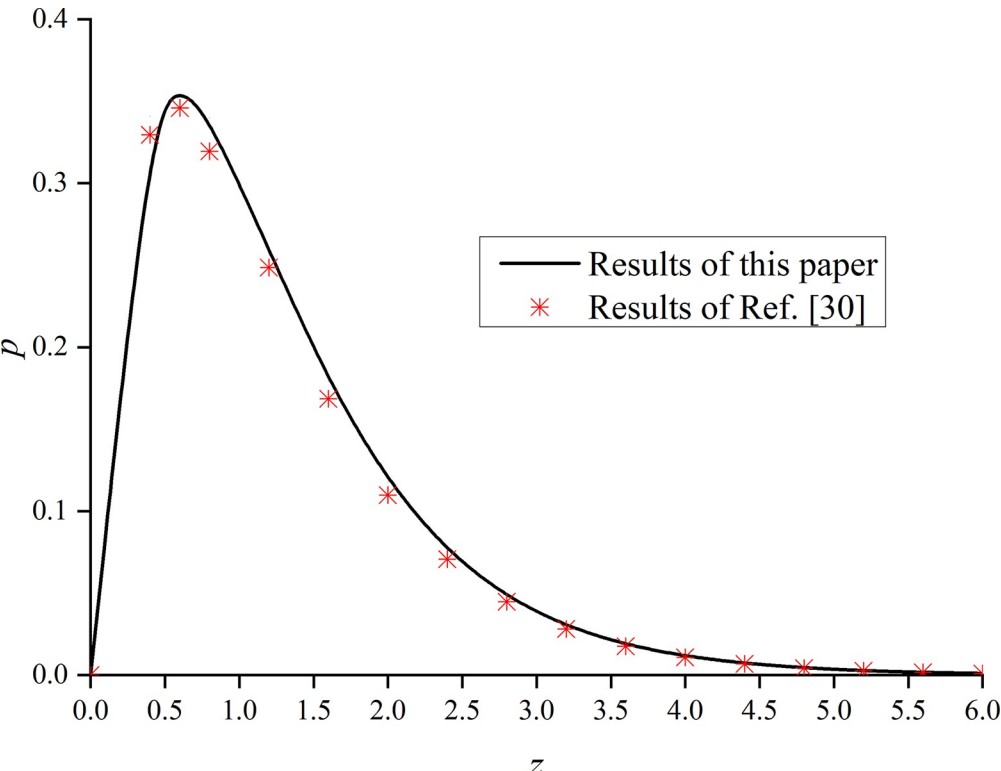

**Fig 3. Comparison between the non-dimension excess pore water pressure and that presented in [30].**

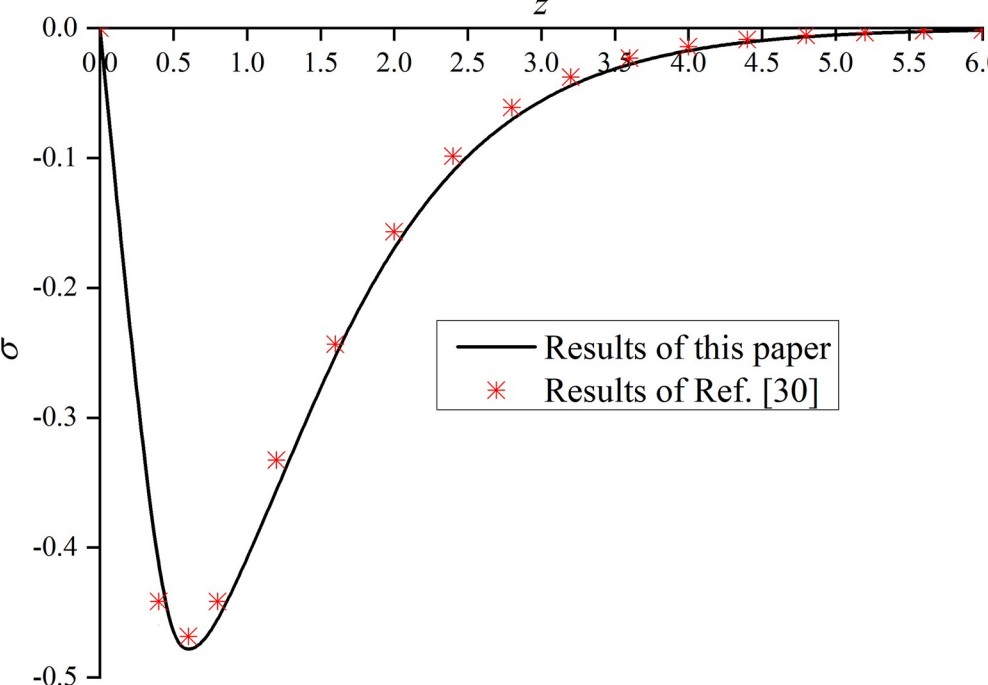

**Fig 4. Comparison between the non-dimension vertical stress and that presented in [30].**

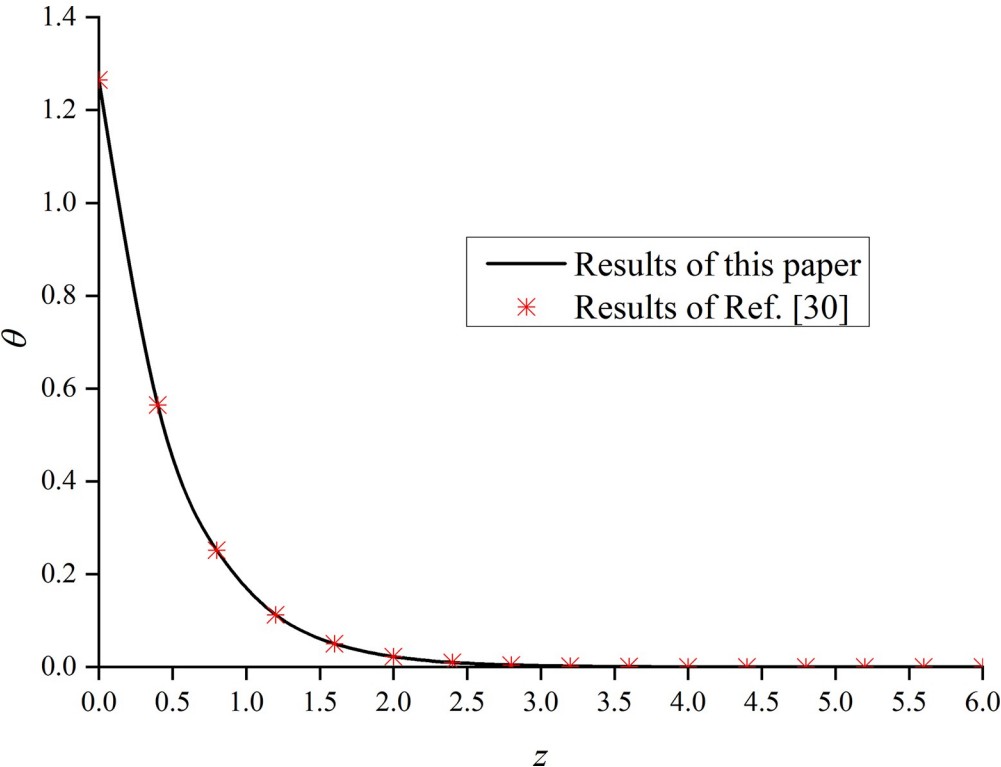

**Fig 5. Comparison between the non-dimension temperature and that presented in [30].**

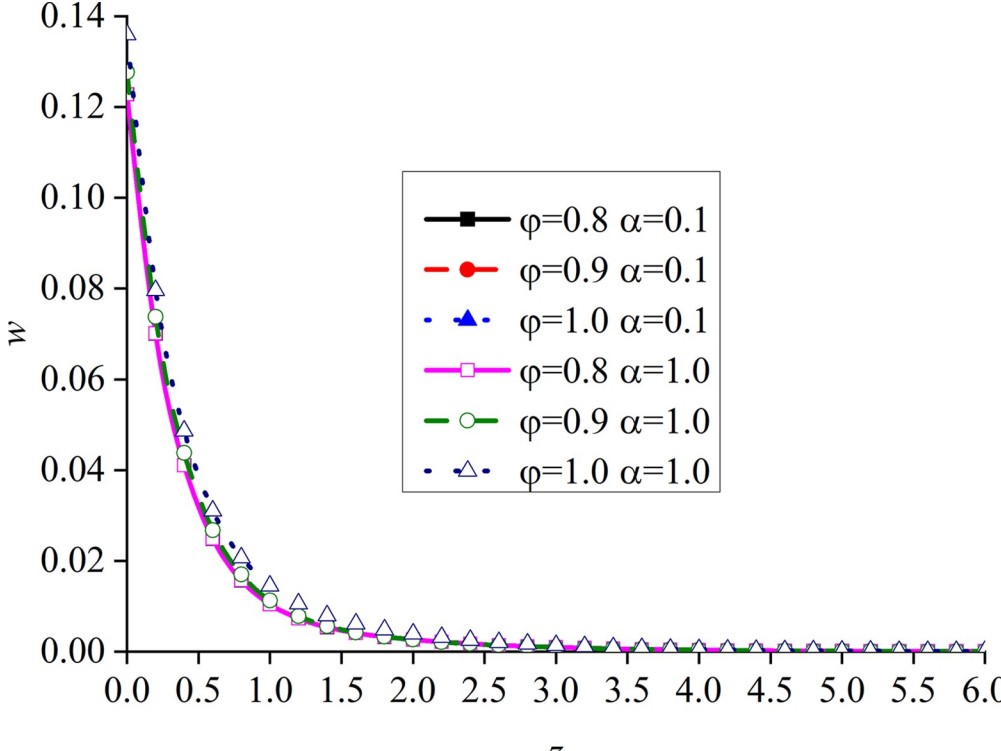

**Fig 6. Distribution of the dimensionless vertical displacement for different values of $\alpha$ and $\varphi$ with mechanical load.**

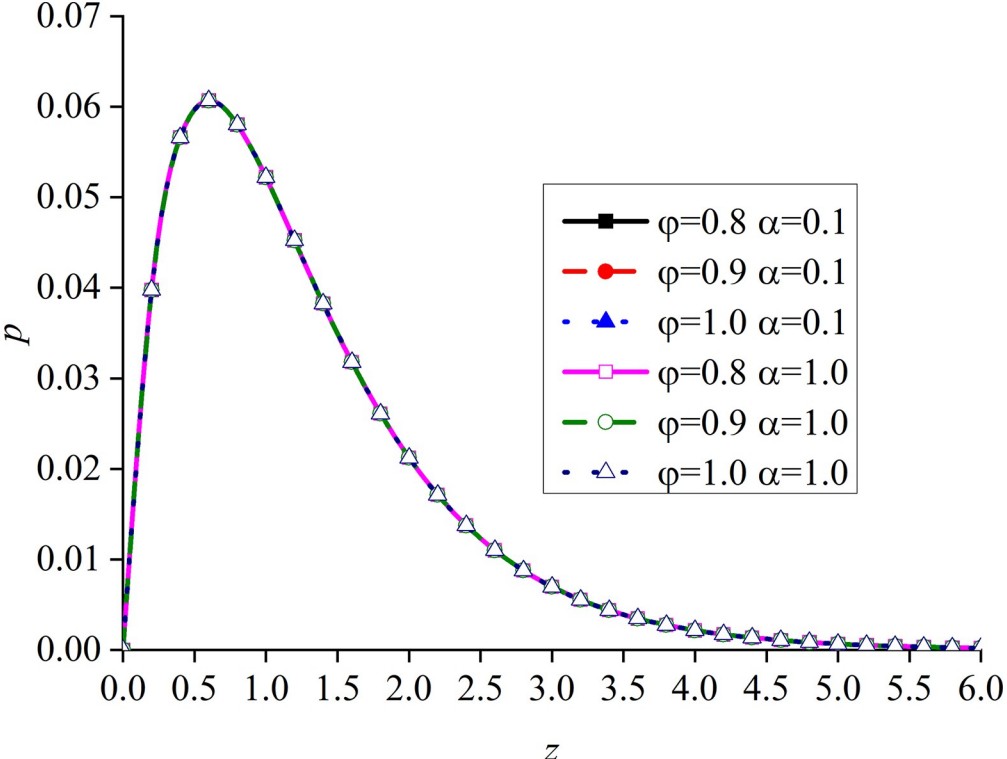

**Fig 7. Distribution of the dimensionless excess pore water pressure for different $\alpha$ and $\varphi$ values with mechanical load.**

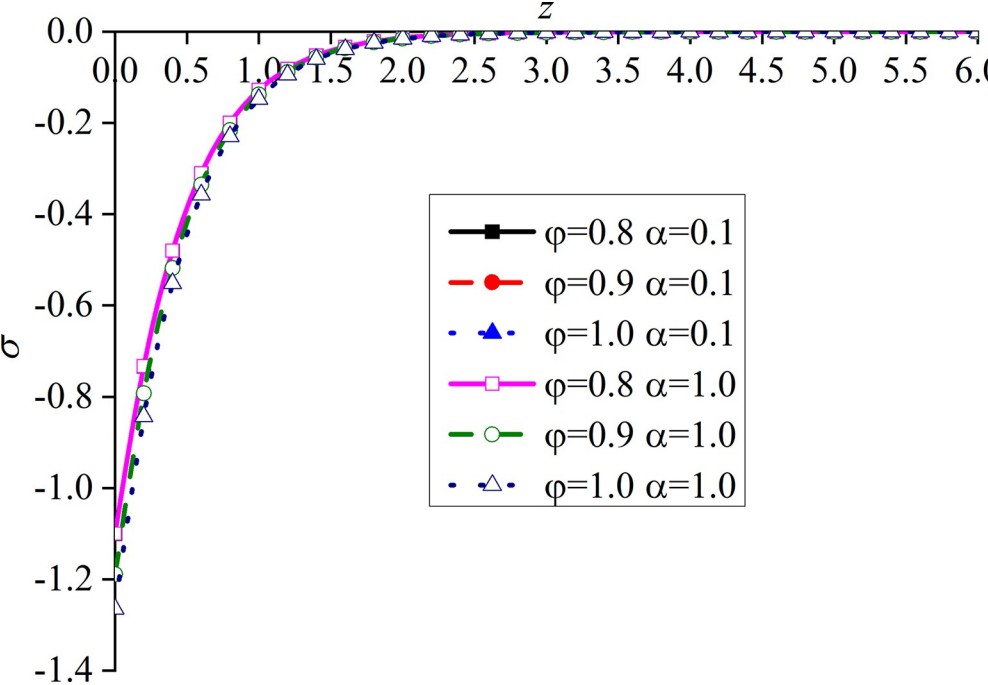

**Fig 8. Distribution of the dimensionless vertical stress for different $\alpha$ and $\varphi$ values with mechanical load.**

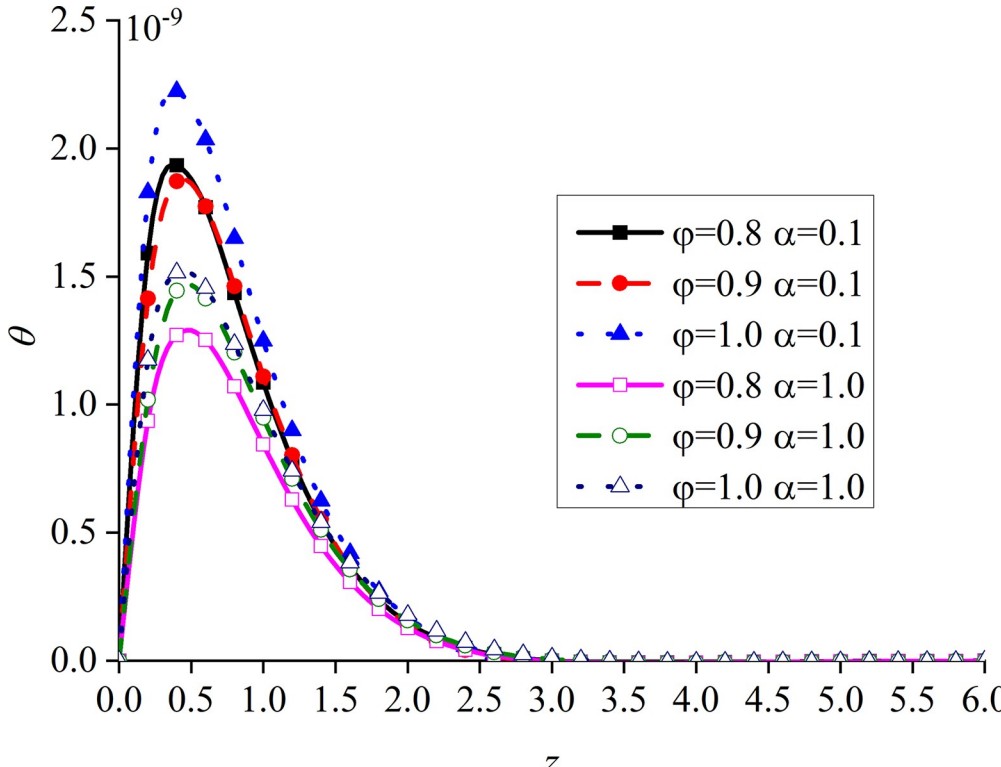

**Fig 9. Distribution of the dimensionless temperature for different $\alpha$ and $\varphi$ values with mechanical load.**

the vertical and horizontal thermal conduction coefficient, while $\varphi$ = 1.0 denotes the isotropy of the heat conduction coefficient, which is consistent with the results presented in [52]. These two sets of figures verify the validity of the proposed model.

In Figs 6 and 8, the dimensionless vertical displacement ($w$) and vertical stress ($\sigma$) are the main differences on the ground surface. When $\varphi$ increases, the curve of these dimensionless variables quickly decreases. This is mainly due to the near-surface of the vertical displacement. In addition, the vertical stress also increases with the increase of $\varphi$. However, at the same depth, all the basic disturbances of the curves disappear. It can be seen from the vertical displacement curve that, when $\varphi$ increases, the difference between the two adjacent curves gradually increases, while the difference between the vertical stress curves remains the same. In addition, in these two figures, the distribution law and disturbance depth of the curves are similar. This is due to the fact that the dimensionless vertical displacement is mainly caused by dimensionless vertical stress under mechanical load, and the influence of the dimensionless pore water pressure is weak. The dimensionless temperature generated by the mechanical load and the dimensionless temperature are very small.

It can be observed from Fig 7 that the change of $\varphi$ and $\alpha$ has no clear effect on $p$. Since the boundary conditions assume that the excess pore water pressure is continuous, it is zero at the boundary. When the depth increases, the excess pore water pressure first increases and then decreases under the action of mechanical load.

It can be seen from Fig 9 that $\varphi$ and $\alpha$ have obvious effects on the dimensionless temperature. When $\alpha$ is small, the curve of $\varphi$ = 1.0 is larger than the other two curves, the curves of $\varphi$ = 0.8 and $\varphi$ = 0.9 are close, while that of $\varphi$ = 0.8 is slightly larger. When $\alpha$ = 1, the curves of $\varphi$ =

0.9 and $\varphi = 1.0$ are close, and the dimensionless temperature curve gradually increases with the increase of $\varphi$. When $\alpha$ increases, the influence of the change of $\varphi$ on the dimensionless temperature becomes more regular. It can be clearly seen that the influence of the change of $\varphi$ on all the physical variables in the subgrade only appears at a certain depth. This further illustrates the importance of the fractional order parameter presented in this study.

Figs 10–13 show the impacts of $\varphi$ and $\alpha$ on four different dimensionless physical variables, after the subgrade surface is subjected to thermal load from the vertical direction. It can be seen that all the dimensionless variables gradually increase with the increase of $\varphi$ and $\alpha$.

It can be observed from Fig 10 that the impact of $\varphi$ on the dimensionless vertical displacement mainly appears at the peak of the curve in the expansion region. The dimensionless vertical displacement generated by the thermal source starts from the compression and moves into the expansion region.

It can be observed from Fig 11 that, when $\alpha$ is small, the impact of $\varphi$ on the dimensionless excess pore water pressure appears at the main peak of the curve. More precisely, the surface of the subgrade and the surrounding area are completely overlapped. When $\alpha$ increases, the difference between the three curves gradually becomes clear after the curve reaches its peak. The upper surface of the subgrade is still completely overlapped, which is caused by the drainage of the upper surface.

It can be seen from Fig 12 that the variation of $\varphi$ makes the difference between the dimensionless vertical stress curves more obvious, which is clearer at the peak of the curves.

The dimensionless temperature mainly overlaps on the subgrade surface, as shown in Fig 13. When the subgrade depth slowly increases, the higher the values of $\varphi$ and $\alpha$, the lower the attenuation rate of the curve, and the difference becomes the most obvious within a certain depth interval of the subgrade. The dimensionless temperature gradually increases with the increase of $\varphi$, which is consistent with the variation law of presented in [52]. This further proves that the developed model is reasonable and the calculated results are correct.

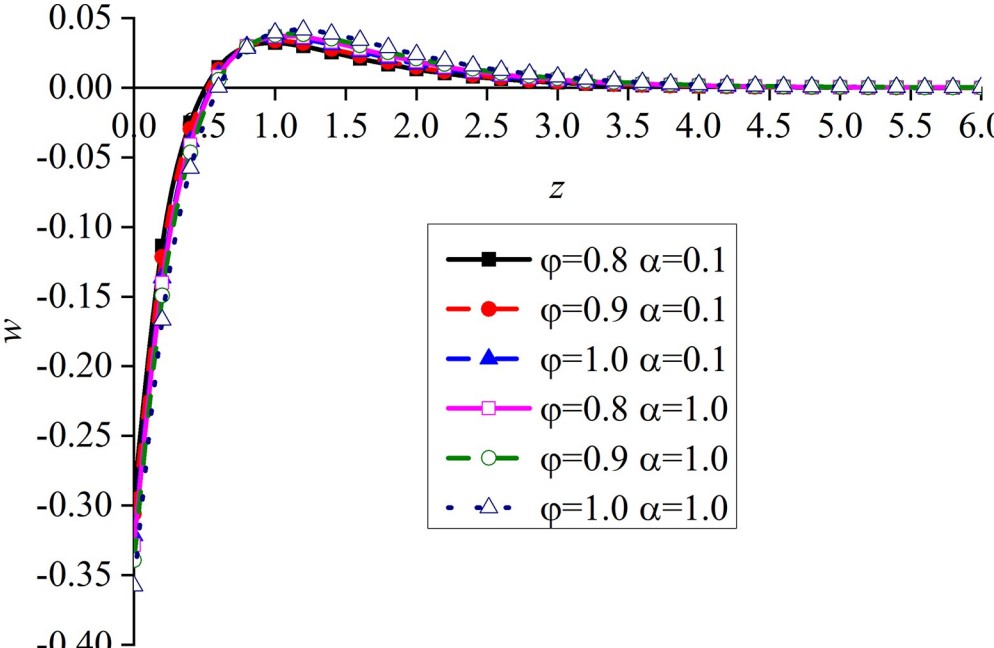

**Fig 10. Distribution of the dimensionless vertical displacement for different values of $\alpha$ and $\varphi$ with thermal source.**

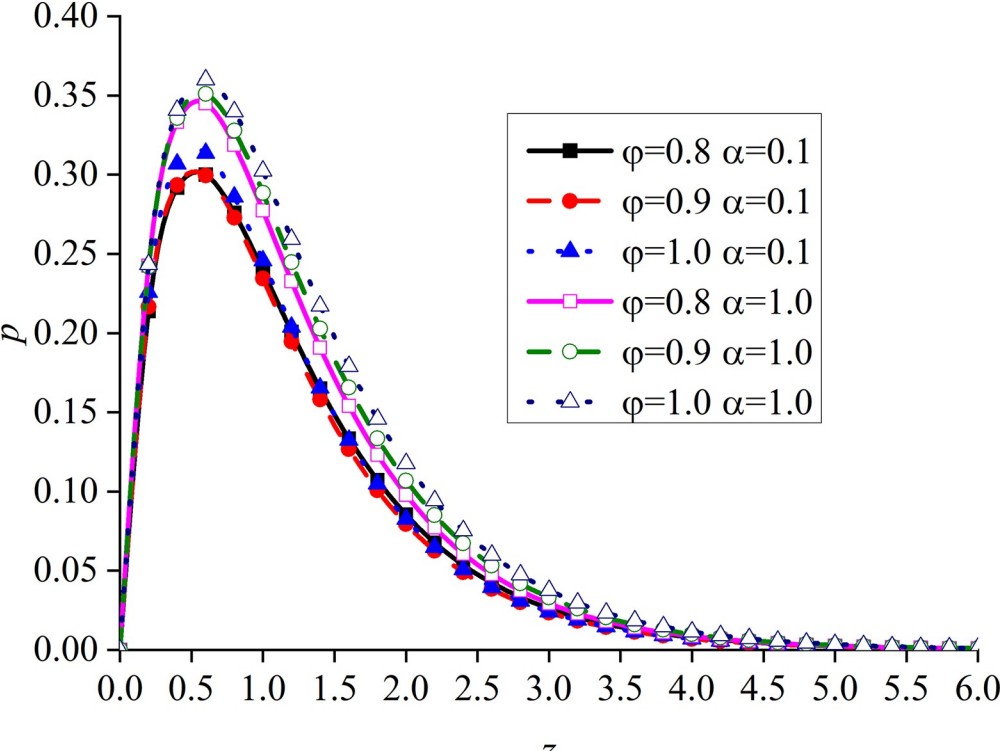

**Fig 11. Distribution of the dimensionless excess pore water pressure for different values of $\alpha$ and $\varphi$ with thermal source.**

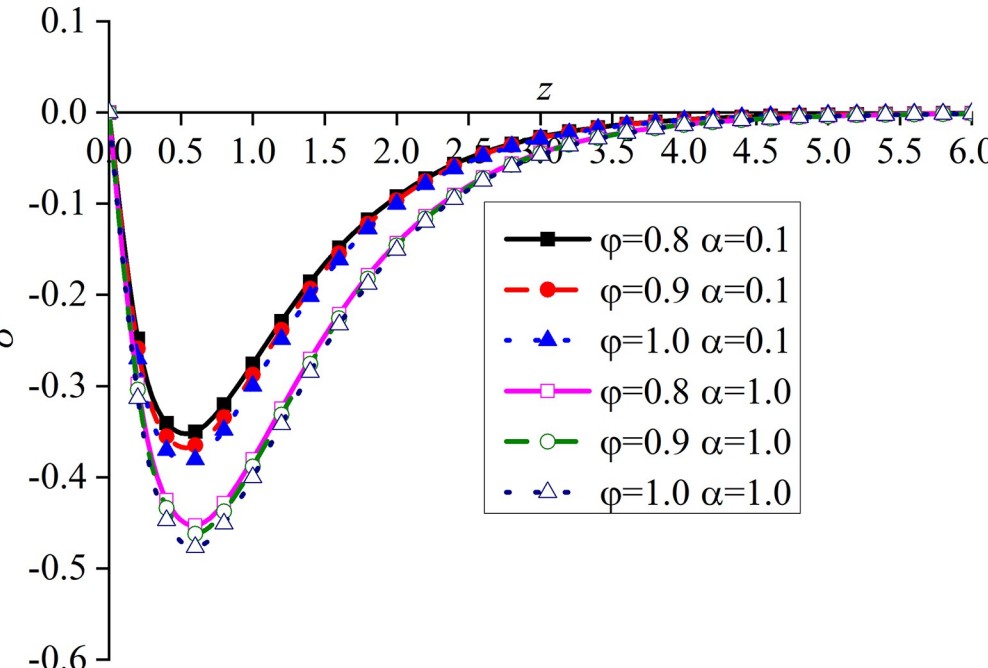

**Fig 12. Distribution of the dimensionless vertical stress for different values of $\alpha$ and $\varphi$ with thermal source.**

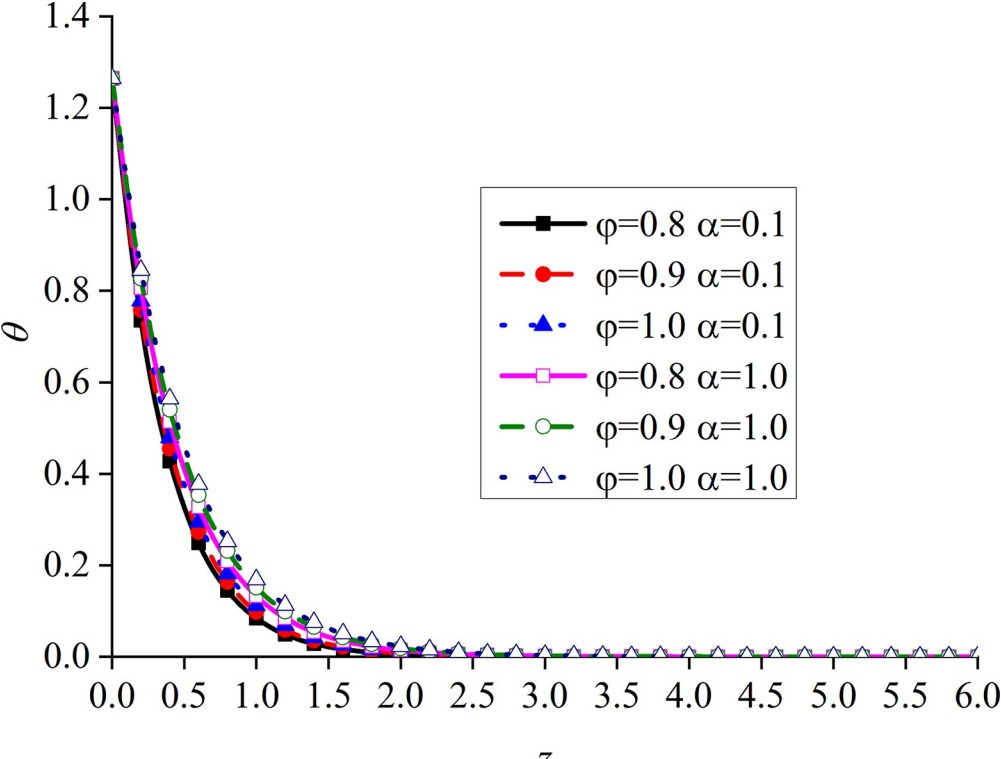

**Fig 13. Distribution of the dimensionless temperature for different values of $\alpha$ and $\varphi$ with thermal source.**

In summary, the influence of the anisotropic parameters of the thermal conductivity coefficient on all the physical variables appears after the subgrade reaches a certain depth. This has a non-negligible impact on the safety and stability of the buried structures, pipelines, and structures.

## 6.2 Effect of the frequency and fractional order parameter with mechanic load or thermal source

Figs 14–17 show the distributions of the dimensionless vertical displacement, excess pore water pressure, vertical stress, and temperature at frequencies of $\omega = 1.6$ and $\omega = 2.5$, under the THMD coupling model. Different $\alpha$ values ($\alpha = 0.1$, $\alpha = 0.5$, and $\alpha = 1.0$) are also considered. Similar to section 6.1, these are selected at $x = 1.0$ and observed during the action time $t = 0.5$. Note that the punctuation is omitted from the Figures.

Figs 14–17 show the distribution of four dimensionless physical variables increasing along the subgrade depth through the change of $\omega$ and $\alpha$. The change of the fractional order parameters in Figs 14–16 has no obvious influence on these three dimensionless physical variables. The vertical displacement and vertical stress curves of the two different frequencies in Figs 14 and 16 intersect at a point. Before this point, the response curves of the two different physical variables increase with the increase of $\omega$. After this intersection point, when $\omega$ increases, the response curves of the two different physical variables decrease. This is because the larger the value of $\omega$, the higher the decay rate of the physical variables. It can be observed from Fig 15 that the increase of $\omega$ significantly increases the excess pore water pressure, which is mainly due to the fact that the pore water has not been discharged in time after the increase of $\omega$.

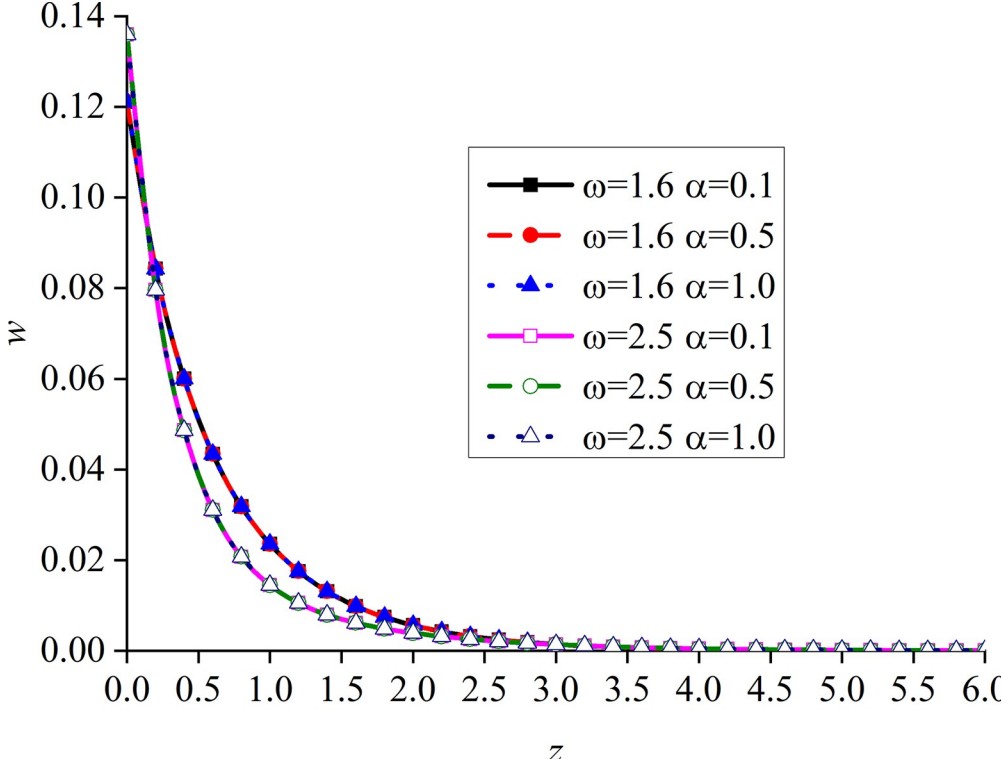

**Fig 14. Impacts of $\alpha$ and $\omega$ on the dimensionless vertical displacement with mechanical load.**

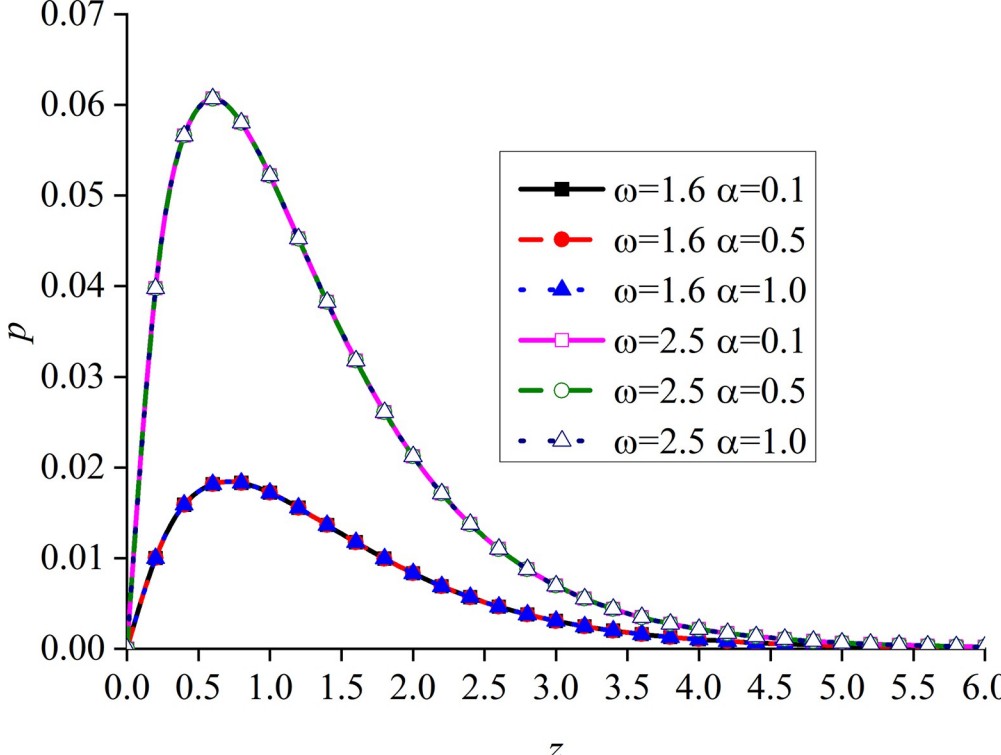

**Fig 15. Impacts of $\alpha$ and $\omega$ on the dimensionless excess pore water pressure with mechanical load.**

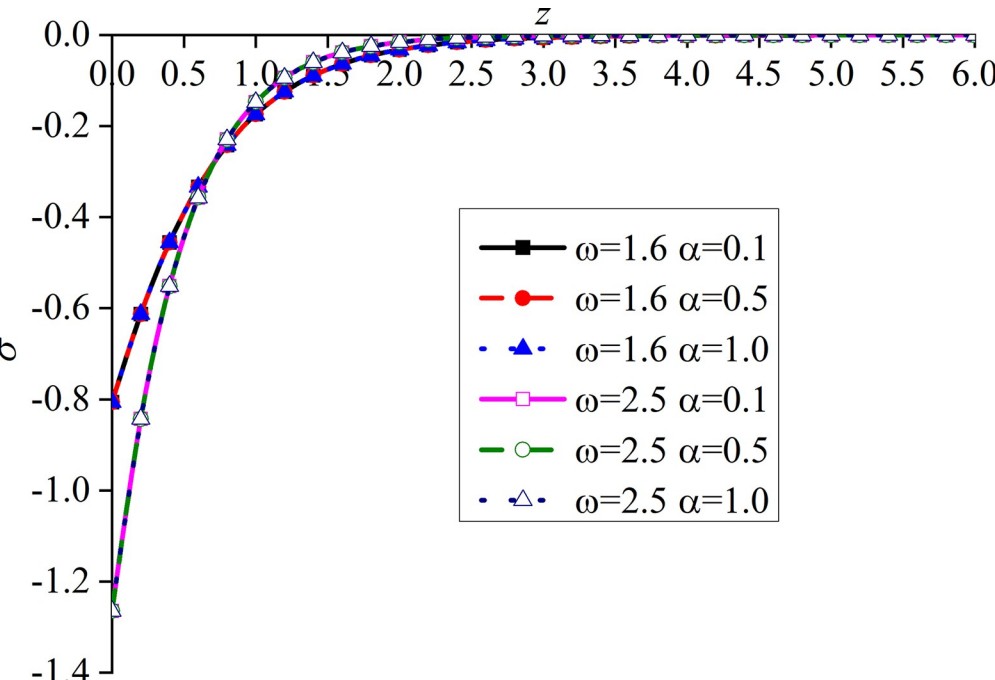

**Fig 16. Impacts of *α* and *ω* on the dimensionless vertical stress with mechanical load.**

It can be seen from Fig 17 that the value of the dimensionless temperature always starts at zero and then decreases along the z-axis. This is due to the fact that the surface is subjected to mechanic load, and the mechanical wave has finite speed propagating in the media. When $\omega$ increases, the curve of $\theta$ also increases, and the peak value is the most obvious, which basically appears at the same subgrade depth. When $\omega$ is constant, $\theta$ gradually increases with the decrease of $\alpha$, and the difference between the two adjacent curves clearly increases. When $\omega$ increases, the peak value of the curve with a larger $\omega$ is larger than that with a smaller one. In addition, the six curves intersect at one point in the attenuation stage, and the curve with larger $\omega$ attenuates faster. It can be seen that the curves with large load frequency increase and decay faster. The influence of $\alpha$ on the curve is mainly at the peak.

The six curves in Figs 18–21 show the relationship between the four different dimensionless physical variables and the change of the subgrade depth when the upper surface is affected by the thermal source.

It can be seen from Fig 18 that the dimensionless vertical displacement increases (from a negative value to a positive one), reaches a peak value, and then gradually decreases until it becomes null. The six curves successively enter the expansion state within a depth range of $0.5 \leq z \leq 1$. When $\omega$ is small, the difference between the vertical displacements caused by the change of $\alpha$ is not obvious. When $\omega$ increases, the difference between the values of $\alpha$ becomes more obvious, which is mainly reflected in the upper surface and the peak of the curve. When $\alpha$ increases, the dimensionless vertical displacements gradually increase. When $\omega$ increases, the difference caused by the change of $\alpha$ becomes more obvious.

It can be seen from Figs 19 and 20 that $\omega$ and $\alpha$ significantly affect the dimensionless excess pore water pressure and vertical stress. When $\omega$ is constant, the increase of $\alpha$ causes the peak values of the curves of the two physical variables to gradually move deeper into the subgrade. When $\omega$ is small, the two curves of $\alpha = 0.5$ and $\alpha = 1.0$ are very close. When $\omega$ increases, the

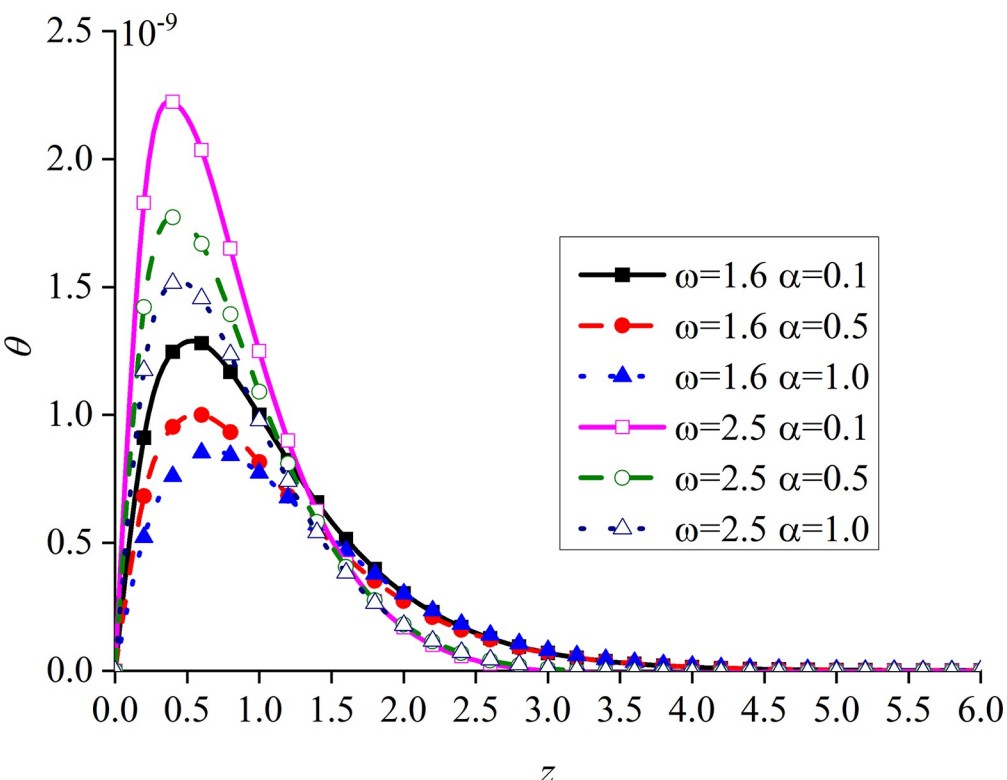

**Fig 17. Impacts of $\alpha$ and $\omega$ on the dimensionless temperature with mechanical load.**

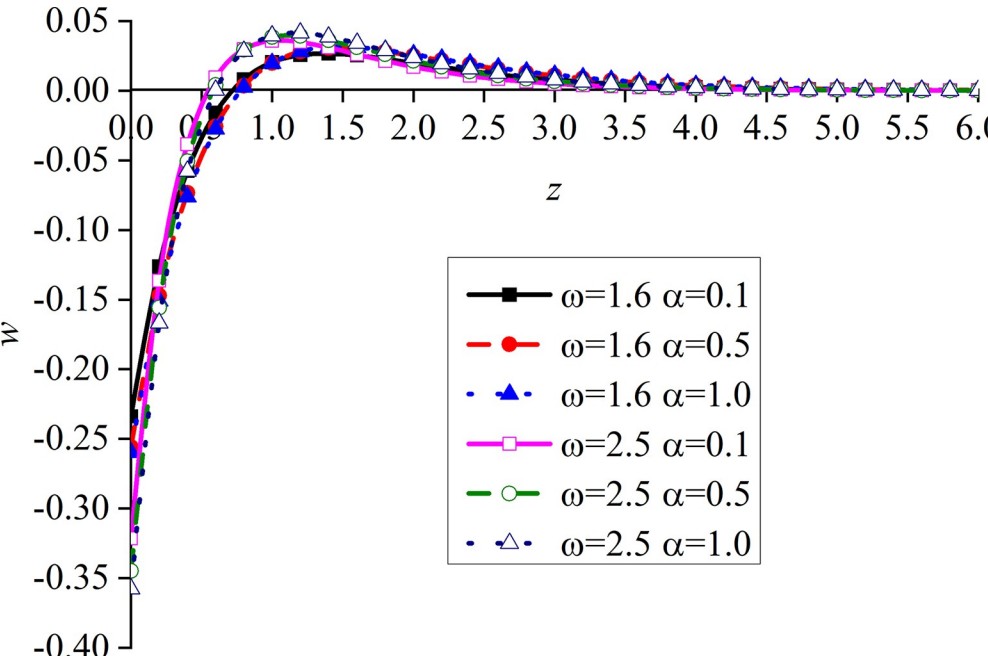

**Fig 18. Impacts of $\alpha$ and $\omega$ on the dimensionless vertical displacement with thermal source.**

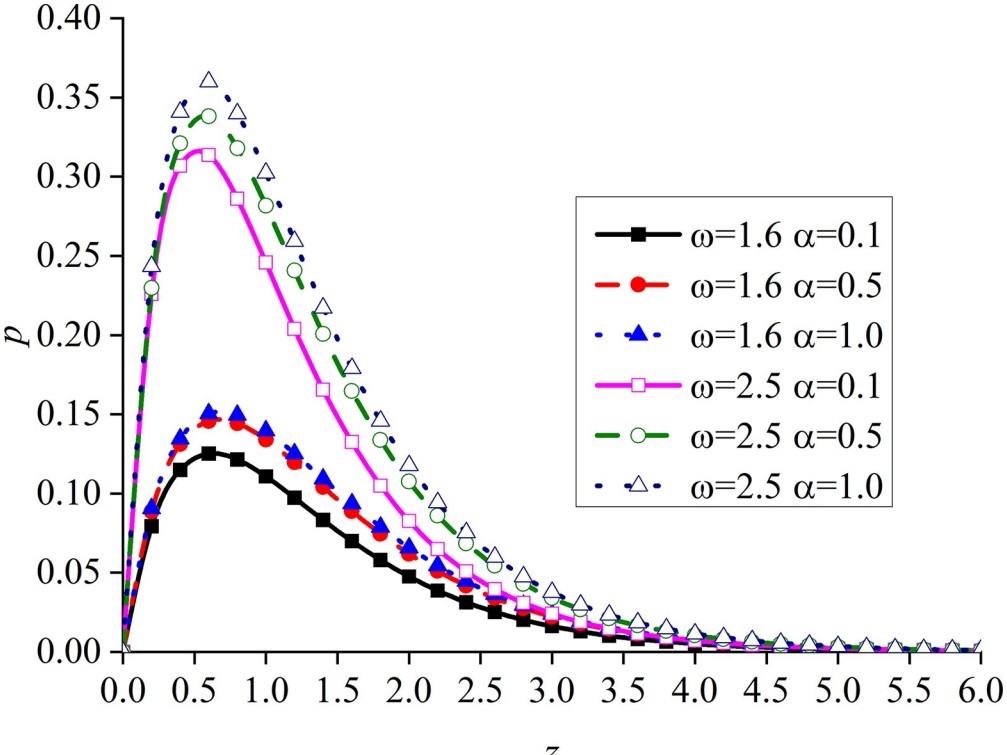

**Fig 19. Impacts of $\alpha$ and $\omega$ on the dimensionless excess pore water pressure with thermal source.**

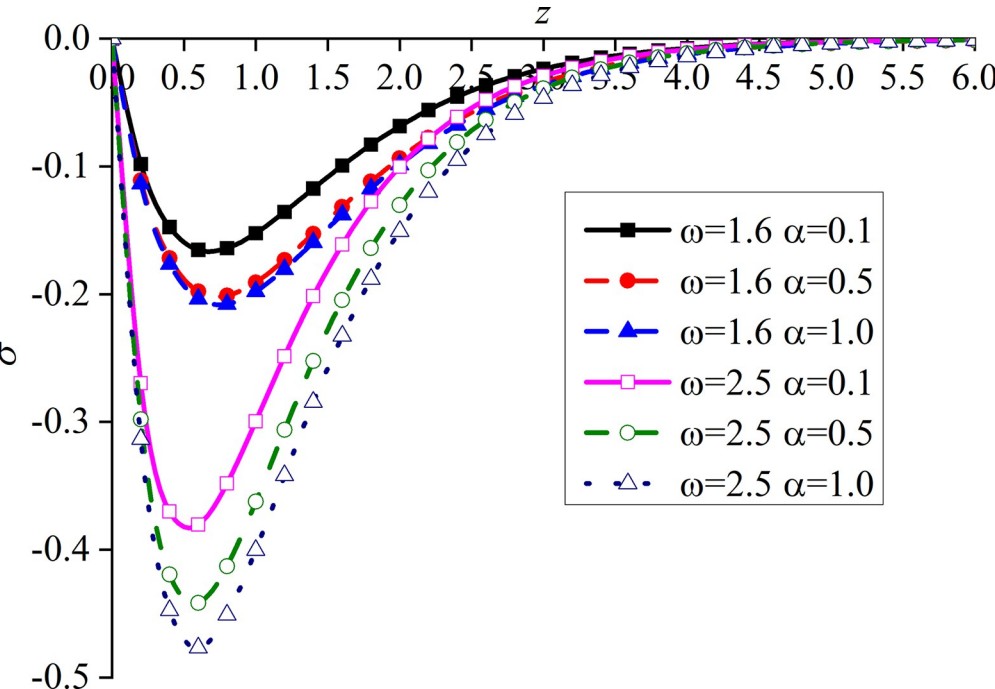

**Fig 20. Impacts of $\alpha$ and $\omega$ on the dimensionless vertical stress with thermal source.**

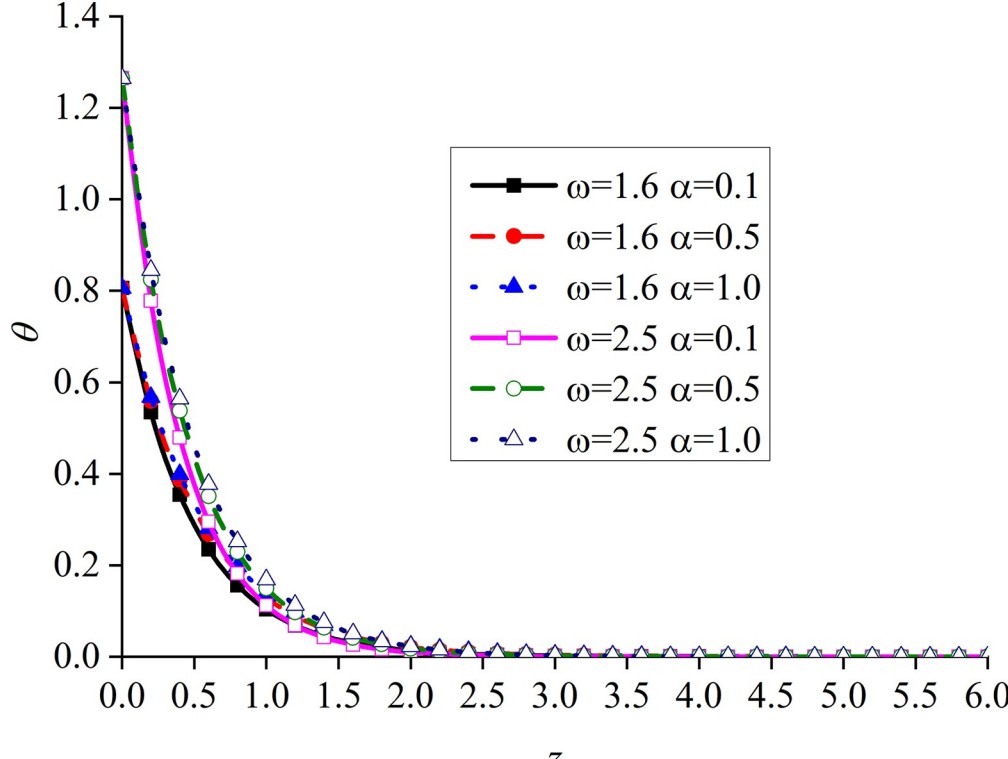

**Fig 21. Impacts of $\alpha$ and $\omega$ on the dimensionless temperature with thermal source.**

difference between the two curves with $\alpha = 0.5$ and $\alpha = 1.0$ significantly increases. In addition, when $\alpha$ increases, the difference between the two adjacent curves gradually decreases. Moreover, the influence of $\alpha$ on the vertical stress and excess pore water pressure is related to the subgrade depth, to a certain extent. In the case of a constant $\omega$, the difference between the three curves is not obvious when they are close to the upper subgrade surface. When the depth reaches $z = 0.5$, the difference is very obvious. In addition, the curve attenuates to zero when the depth is $z = 4.5$.

It can be seen from Fig 21 that $\alpha$ has a certain influence on the dimensionless temperature. More precisely, when $w$ is constant, the temperature curves at the surface with different $\alpha$ values are close to each other. When the subgrade depth increases, the variation of $\alpha$ with the change of the dimensionless temperature becomes more obvious. In addition, when the subgrade depth increases, all the curves monotonically decrease and reach zero at a depth of $z = 2$. When $\omega$ is constant, the attenuation rate of the curve with small $\alpha$ value is slightly faster than that with large $\alpha$ value. The higher the value of $\omega$, the more obvious the influence of the fractional coefficients on the dimensionless temperature.

## 7. Conclusion

This paper studies the impact of different variables on the dimensionless vertical displacement, excess pore water pressure, vertical stress, and temperature of the 2D anisotropic fully fluid-saturated semi-infinite subgrade under mechanical load or thermal source, based on the Ezzat's fractional order generalized theory of thermoelasticity, which considers the Riemann-Liouville integral operator and Darcy's law. The impacts of the load frequency, fractional order parameter, and anisotropy of thermal conduction coefficient on different physical variables in

the subgrade are detailed. A normal mode analysis method is used to solve this THMD coupling problem. The conclusions drawn from the obtained results can be summarized as follows:

1. The normal mode analysis method is successfully applied to the solution of the eighth-order characteristic equation of anisotropic media. It can accurately analyze all the considered distributions of physical variables.

2. The frequency plays a crucial role for all the dimensionless physical variables regardless of the load type. In particular, it affects the distribution of various physical variables when considering the influence of thermal loads.

3. The fractional order coefficient has only a clear effect on the dimensionless temperature when the mechanical load is considered on the upper surface of the subgrade. However, it has obvious effect on the dimensionless excess pore water pressure, vertical stress, and temperature when the upper surface of the subgrade is subjected to thermal load. The variation of the load frequency will affect the fractional coefficient on physical variables.

4. The variation of the anisotropy of thermal conduction coefficient significantly affects all the considered dimensionless variables. When the anisotropy of thermal conduction coefficient increases, all the physical variables gradually increase. The existence of fractional order parameters makes this effect more obvious, especially in the dimensionless temperature curve under mechanical load, the dimensionless excess pore water pressure curve, and the vertical stress curve under thermal source.

## Supporting information

**S1 Data.**
(ZIP)

## Author Contributions

**Conceptualization:** Ying Guo.

**Data curation:** Jie Li.

**Formal analysis:** Ying Guo.

**Funding acquisition:** Ying Guo, Chunbao Xiong.

**Investigation:** Cui Du.

**Methodology:** Ying Guo, Wen Yu.

**Supervision:** Ying Guo, Chunbao Xiong.

**Validation:** Wen Yu, Cui Du.

**Writing – original draft:** Ying Guo.

**Writing – review & editing:** Jianjun Ma.

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
