## [Decision Letter · Decision Letter 0]

9 Oct 2023

PONE-D-23-25844Coupling Dynamic Response of Saturated Soil with Anisotropic Thermal Conductivity under Fractional Order Thermoelastic TheoryPLOS ONE

Dear Dr. Yu,

Thank you for submitting your manuscript to PLOS ONE. After careful consideration, we feel that it has merit but does not fully meet PLOS ONE’s publication criteria as it currently stands. Therefore, we invite you to submit a revised version of the manuscript that addresses the points raised during the review process.

We look forward to receiving your revised manuscript.

Kind regards,

Ghulam Rasool

Academic Editor

PLOS ONE

Journal Requirements:

 "This work was supported in part by Natural Science Foundation of Henan Province (222300420153), Natural Science Foundation of Tianjin (S22YBJ1114), State Key Laboratory of Hydraulic Engineering Simulation and Safety, Tianjin University (HESS-2324), 2022 Heluo Young Talent Lifting Project (2022HLTJZC10), Scientific and Technological Project in Henan Province (212102310055), and Key Scientific Research Project of Henan Province (22A130004)."

"This work was supported in part by Natural Science Foundation of Henan Province

(222300420153), Natural Science Foundation of Tianjin (S22YBJ1114), State Key

Laboratory of Hydraulic Engineering Simulation and Safety, Tianjin University

(HESS-2324), 2022 Heluo Young Talent Lifting Project (2022HLTJZC10), Scientific

and Technological Project in Henan Province (212102310055), and Key Scientific

Research Project of Henan Province (22A130004)."

 "This work was supported in part by Natural Science Foundation of Henan Province (222300420153), Natural Science Foundation of Tianjin (S22YBJ1114), State Key Laboratory of Hydraulic Engineering Simulation and Safety, Tianjin University (HESS-2324), 2022 Heluo Young Talent Lifting Project (2022HLTJZC10), Scientific and Technological Project in Henan Province (212102310055), and Key Scientific Research Project of Henan Province (22A130004)."

**Additional Editor Comments:**

We have received some suggestions to revise the current form of paper. The suggested comments of reviewers are provided in review reports. Please incorporate the comments one by one and provide a detailed answer sheet during the time of revision submission.

Reviewers' comments:

Reviewer's Responses to Questions

**Comments to the Author**

1. Is the manuscript technically sound, and do the data support the conclusions?

Reviewer #1: Partly

Reviewer #2: Yes

2. Has the statistical analysis been performed appropriately and rigorously? 

Reviewer #1: Yes

Reviewer #2: N/A

3. Have the authors made all data underlying the findings in their manuscript fully available?

Reviewer #1: Yes

Reviewer #2: Yes

4. Is the manuscript presented in an intelligible fashion and written in standard English?

Reviewer #1: Yes

Reviewer #2: Yes

5. Review Comments to the Author

Reviewer #1: The research article titled "Coupling Dynamic Response of Saturated Soil with Anisotropic Thermal Conductivity under Fractional Order Thermoelastic Theory" investigated a two-dimensional (2D) thermo-hydro-mechanical dynamic (THMD) coupling analysis in the presence of a half-space medium by using Ezzat’s fractional order generalized theory of thermoelastic. In this paper, the general relationships among the dimensionless physical variables including vertical displacement, excess pore water pressure, vertical stress, and temperature distribution was discussed. The calculated results can guide the geotechnical engineering construction according to different values of load frequency, fractional order coefficient, anisotropy of thermal conduction coefficient which improves the stability of subgrade and enriches the theoretical research in the field of thermo-hydro-mechanical coupling.

The paper is well-written, the novelty is clearly emphasized and can be accepted after the suggested minor revisions. Please provide a clear and concise explanation of the motivations and objectives of the study in the Abstract which is clearly missing. It is better to do some more comparison of the work with other relevant work with proper references. It is highly suggested to include numerical values in the conclusions for the better understanding which is also missing. Please check all the references format and is suggested to add some of the very latest work citations from 2023.

Reviewer #2: I suggest the authors to consider the following encouraging points to further improve the paper presentation and content;

1. **Introduction:**

Revisit the introduction section to provide a more comprehensive background on Fractional Order Thermoelastic Theory application.

2. **Methodology:**

Expand the methodology section to include a detailed description of the numerical approach, numerical methods, and computational tools used in the study.

3. **Model Validation:**

Include a robust validation section that demonstrates the accuracy and reliability of the numerical model. Compare computational results with relevant experimental or clinical data to establish the model's credibility.

4. **Discussion:**

Expand the discussion section to provide more nuanced interpretations of the results. Address the advantages and limitations of Normal mode analysis. Explore potential clinical implications and future research directions.

5. **Conclusion:**

Summarize the major findings and the broader implications of the enhanced study. Highlight the significance of the refined numerical approach.

6. **Language and Clarity:**

Review the paper for language clarity, grammar, and overall readability. Ensure that the flow of the paper is logical and coherent, with smooth transitions between sections.

6. PLOS authors have the option to publish the peer review history of their article (what does this mean?). If published, this will include your full peer review and any attached files.

Reviewer #1: No

Reviewer #2: **Yes: **Hasan Shahzad

---

## [Author Response · Author response to Decision Letter 0]

22 Nov 2023

Responses to reviewers’ comments

Dear Reviewers,

Thank you for your consideration. In addition, we sincerely thank all of you for your valuable comments, which helped us in the revision and enhancement of the presentation of our manuscript.

Following the suggestions provided by the reviewers, we revised our manuscript point by point carefully. All changes have been highlighted in blue in the manuscript and are enclosed in this file. If there are any questions, please let us know, and we will try our best to further improve the quality of our manuscript.

Best wishes,

Ying Guo

Comments and revision

Reviewer #1: 

The research article titled "Coupling Dynamic Response of Saturated Soil with Anisotropic Thermal Conductivity under Fractional Order Thermoelastic Theory" investigated a two-dimensional (2D) thermo-hydro-mechanical dynamic (THMD) coupling analysis in the presence of a half-space medium by using Ezzat’s fractional order generalized theory of thermoelastic. In this paper, the general relationships among the dimensionless physical variables including vertical displacement, excess pore water pressure, vertical stress, and temperature distribution was discussed. The calculated results can guide the geotechnical engineering construction according to different values of load frequency, fractional order coefficient, anisotropy of thermal conduction coefficient which improves the stability of subgrade and enriches the theoretical research in the field of thermo-hydro-mechanical coupling.

The paper is well-written, the novelty is clearly emphasized and can be accepted after the suggested minor revisions. Please provide a clear and concise explanation of the motivations and objectives of the study in the Abstract which is clearly missing. It is better to do some more comparison of the work with other relevant work with proper references. It is highly suggested to include numerical values in the conclusions for the better understanding which is also missing. Please check all the references format and is suggested to add some of the very latest work citations from 2023.

Response: 

We added some new references to provide a more comprehensive background of our investigated. The added parts are highlighted in blue on page 4 (lines 24-25), page 5 (lines 30-34), page 6 (lines 1-5, 8-18), page 25 (lines 4-12), page 27 (lines 12-25, 29-30), and page 28 (lines 1-10), read as follows:

“These theories have been widely used to solve many kinds of problems [10-17].

Zhang and Ma [41] studied the nonlocal fractional order strain problem which is subject to moving heat source using the fractional order strain theory. Abouelregal et al. [42] proposed a generalized thermoelastic model with two-temperature characteristics: the heat transfer equation with Caputo-Fabrizio fractional differential operator and the phase lags. Abouelregal et al. [43] analyzed the thermoelastic vibrations of a nonlocal isotropic solid medium subjected to a pulsed heat flux based on the Caputo-Fabrizio fractional derivative generalized thermoelasticity. AL-Lehaibi [44] solved a two-dimensions thermoelastic problem for an isotropic and homogeneous nanobeam in terms of the thermoelasticity with one relaxation time and fractional-order strain theory. 

Hu et al. [46] used a fractional dual-phase-lag bio-thermoelastic model to solve the thermoelastic response of a biological tissue during hyperthermia treatment by a moving laser heating. Han et al. [47] studied the thermoelastic transient response of porous microplates subjected to thermal and stress shocks at the left boundary based on the Atangana-Baleanu fractional order generalized thermoelastic theory. This was performed by considering nonlocal effects and fractional order strain combined with the thermoelasticity theory of porous materials and the dual phase-lag heat conduction model. Dutta et al. [48] proposed a generalized thermo-diffusion process in a semi-infinite nonlocal fiber-reinforced double porous thermoelastic diffusive material with voids using the fractional-order Lord-Shulman thermo-elasto-diffusion theory.

[15]Othman M I A, Mansour N T. Effect of relaxation time on generalized double porosity thermoelastic medium with diffusion. Geomechanics and Engineering, 2023, 32(5): 475-482.

[16]Zhang W L, Zhao Y, Zhao X D, et al. Thermoelastic response of laminated plates considering interfacial conditions and cracks based on peridynamics. Acta Mechanica, 2023, 234: 2179-2203.

[17]Paul K, Mukhopadhyay B. A novel mathematical model on generalized thermoelastic diffusion theory. Journal of Thermal Stresses, 2023, 46(4): 253-275.

[41]Zhang J, Ma Y. Thermoelastic response of an elastic rod under the action of a moving heat source based on fractional order strain theory considering nonlocal effects. International Journal for Computational Methods in Engineering Science and Mechanics, 2023, DOI: 10.1080/15502287.2023.2265357.

[42]Abouelregal A E, Sofiyev A H, Sedighi H M, et al. Generalized heat equation with the Caputo-Fabrizio fractional derivative for a nonsimple thermoelastic cylinder with temperature-dependent properties. Physical Mesomechanics, 2023, 26(2): 224-240.

[43]Abouelregal A E, Akgöz B, Civalek Ö. Nonlocal thermoelastic vibration of a solid medium subjected to a pulsed heat flux via Caputo-Fabrizio fractional derivative heat conduction. Applied Physics A, 2022, 128: 660.

[44]AL-Lehaibi E. The vibration of a gold nanobeam under the thermoelasticity fractional-order strain theory based on Caputo-Fabrizio’s definition. Journal of Strain Analysis, 2023, 58(6): 464-474.

[46]Hu Y, Zhang X Y, Li X T. Thermoelastic analysis of biological tissue during hyperthermia treatment for moving laser heating using fractional dual-phase-lag bioheat conduction. International Journal of Thermal Sciences, 2022, 182: 107806.

[47]Han Y M, Tian L C, He T H. Investigation on the thermoelastic response of a porous microplate in a modified fractional-order heat conduction model incorporating the nonlocal effect. Mechanics of Advanced Materials and Structures, 2023, DOI: 10.1080/15376494.2023.2238215.

[48]Dutta R, Das S, Gupta S, et al. Nonlocal fiber-reinforced double porous material structure under fractional-order heat and mass transfer. International Journal of Numerical Methods for Heat and Fluid Flow, 2023, DOI 10.1108/HFF-05-2023-0295.

”

Reviewer #2: 

Title: " Coupling Dynamic Response of Saturated Soil with Anisotropic Thermal Conductivity under Fractional Order Thermoelastic Theory "

Abstract:

This paper presents a significant revision and expansion of the study titled " Coupling Dynamic Response of Saturated Soil with Anisotropic Thermal Conductivity under Fractional Order Thermoelastic Theory." This article addresses A two-dimensional (2D) thermo-hydro-mechanical dynamic (THMD) coupling analysis in the presence of a half-space medium is investigated by using Ezzat’s fractional order generalized theory of thermoelastic. The influence of the anisotropy of thermal conduction coefficient, fractional derivatives, and frequency on the THMD response of anisotropy, fully saturated, and poroelastic subgrade is then analyzed with a time harmonic load including mechanical load and thermal source subjected to the surface by using normal mode analysis (NMA). The general relationships among the dimensionless physical variables including vertical displacement, excess pore water pressure, vertical stress, and temperature distribution are graphically illustrated. NMA method does not need integration and inverse transformation, improves the decoupling speed, and eliminates the limitation of numerical inverse transformation. The calculated results can guide the geotechnical engineering construction according to different values of load frequency, fractional order coefficient, anisotropy of thermal conduction coefficient which improves the stability of subgrade, and enriches the theoretical research in the field of thermo-hydro-mechanical coupling.

I suggest the authors to consider the following encouraging points to further improve the paper presentation and content;

1. **Introduction:**

Revisit the introduction section to provide a more comprehensive background on Fractional Order Thermoelastic Theory application. 

Response: 

We added some references to provide a more comprehensive background of fractional order thermoelastic theory. The added parts are highlighted in blue on page 5 (lines 30-34), page 6 (lines 1-5, 8-18), page 27 (lines 12-25, 29-30), and page 28 (lines 1-10), read as follows:

“Zhang and Ma [41] studied the nonlocal fractional order strain problem which is subject to moving heat source using the fractional order strain theory. Abouelregal et al. [42] proposed a generalized thermoelastic model with two-temperature characteristics: the heat transfer equation with Caputo-Fabrizio fractional differential operator and the phase lags. Abouelregal et al. [43] analyzed the thermoelastic vibrations of a nonlocal isotropic solid medium subjected to a pulsed heat flux based on the Caputo-Fabrizio fractional derivative generalized thermoelasticity. AL-Lehaibi [44] solved a two-dimensions thermoelastic problem for an isotropic and homogeneous nanobeam in terms of the thermoelasticity with one relaxation time and fractional-order strain theory. 

Hu et al. [46] used a fractional dual-phase-lag bio-thermoelastic model to solve the thermoelastic response of a biological tissue during hyperthermia treatment by a moving laser heating. Han et al. [47] studied the thermoelastic transient response of porous microplates subjected to thermal and stress shocks at the left boundary based on the Atangana-Baleanu fractional order generalized thermoelastic theory. This was performed by considering nonlocal effects and fractional order strain combined with the thermoelasticity theory of porous materials and the dual phase-lag heat conduction model. Dutta et al. [48] proposed a generalized thermo-diffusion process in a semi-infinite nonlocal fiber-reinforced double porous thermoelastic diffusive material with voids using the fractional-order Lord-Shulman thermo-elasto-diffusion theory.

[41]Zhang J, Ma Y. Thermoelastic response of an elastic rod under the action of a moving heat source based on fractional order strain theory considering nonlocal effects. International Journal for Computational Methods in Engineering Science and Mechanics, 2023, DOI: 10.1080/15502287.2023.2265357.

[42]Abouelregal A E, Sofiyev A H, Sedighi H M, et al. Generalized heat equation with the Caputo-Fabrizio fractional derivative for a nonsimple thermoelastic cylinder with temperature-dependent properties. Physical Mesomechanics, 2023, 26(2): 224-240.

[43]Abouelregal A E, Akgöz B, Civalek Ö. Nonlocal thermoelastic vibration of a solid medium subjected to a pulsed heat flux via Caputo-Fabrizio fractional derivative heat conduction. Applied Physics A, 2022, 128: 660.

[44]AL-Lehaibi E. The vibration of a gold nanobeam under the thermoelasticity fractional-order strain theory based on Caputo-Fabrizio’s definition. Journal of Strain Analysis, 2023, 58(6): 464-474.

[46]Hu Y, Zhang X Y, Li X T. Thermoelastic analysis of biological tissue during hyperthermia treatment for moving laser heating using fractional dual-phase-lag bioheat conduction. International Journal of Thermal Sciences, 2022, 182: 107806.

[47]Han Y M, Tian L C, He T H. Investigation on the thermoelastic response of a porous microplate in a modified fractional-order heat conduction model incorporating the nonlocal effect. Mechanics of Advanced Materials and Structures, 2023, DOI: 10.1080/15376494.2023.2238215.

[48]Dutta R, Das S, Gupta S, et al. Nonlocal fiber-reinforced double porous material structure under fractional-order heat and mass transfer. International Journal of Numerical Methods for Heat and Fluid Flow, 2023, DOI 10.1108/HFF-05-2023-0295.

”

2. **Methodology:**

Expand the methodology section to include a detailed description of the numerical approach, numerical methods, and computational tools used in the study.

Response: 

To improve the readability of the article, a statement is added to describe the calculation method more clearly. The added parts are highlighted in blue on page 10 (lines 9-23), read as follows:

“The normal mode analysis is an inexpensive technique which simulates the low and large amplitude motions. It solves the problem of discrete and truncation errors in the numerical inverse transformation, and it can be divided into two parts without integral transformation and inverse transformation, which increases the decoupling speed. In addition, it can be used to characterize the macro-molecular flexibility to predict the directions of compositional changes and interpret the structural experimental data. This method is applied in various fields such as structural engineering, biological physics, molecular spectroscopy, and structural biology. In this study, the NMA weighted residual method is introduced. The solution of the considered variable can be decomposed into the following normal modes [57]:

 (17)

where frequency is a complex time constant, is the imaginary value, a is the number of waves along the -direction, and represents the amplitudes of the field variables. Note that the value of is related to the wave number, wavelength, and frequency. 

”

3. **Model Validation:**

Include a robust validation section that demonstrates the accuracy and reliability of the numerical model. Compare computational results with relevant experimental or clinical data to establish the model's credibility.

Response: 

We added four figures to verify the rationality of the model. The added parts are highlighted in blue on page 15 (lines 5-21), read as follows (The figures in the text are placed in a separate file):

“

Fig 2. Comparison between the non-dimension displacement and that presented in [30]

Fig 3. Comparison between the non-dimension excess pore water pressure and that presented in [30]

Fig 4. Comparison between the non-dimension vertical stress and that presented in [30]

Fig 5. Comparison between the non-dimension temperature and that presented in [30]

In this paper, and are set to 1. That is, the model is degraded to the dynamic response problem of isotropic media under the integer order generalized thermoelastic theory. In addition, the problem is solved using the boundary conditions presented in [30], which results are compared with the obtained dimensionless displacement, excess pore water pressure, vertical stress, and temperature under thermal shock. It can be seen from Figs. 2-4 that, for and , the results of the two studies are mainly consistent, and the different numerical calculations cause small errors, with maximum and average deviations of 5.07% and 2.5%, respectively. The correctness and rationality of the calculated results are further verified.”

4. **Discussion:**

Expand the discussion section to provide more nuanced interpretations of the results. Address the advantages and limitations of Normal mode analysis. Explore potential clinical implications and future research directions.

Response: 

Normal mode analysis is a method to derive analytical solutions by using weighted residuals, which can eliminated the limitation of the discrete error and truncation error in the numerical inverse transformation, and also can be divided into two parts without integral transformation and inverse transformation, thereby increasing the speed of decoupling. This method is not proposed by us, and it has been applied in many fields[A-1~A-6]. We only apply this method to the solve the problem of saturated porous media. This paper aims to use the normal mode analysis to analyze the multi-physical field coupling problem of foundation under the fractional order parameters and frequency, rather than to study the mathematical problem of normal mode analysis. The normal mode analysis aim at look for the solution in the Fourier transformed domain[A-7~A-8]. Frequency really determines the attenuation speed of wave amplitude, and wave number determines the distance of wave propagation. Therefore, we have investigated the effects of anisotropy of heat conductivity, fractional order parameters and frequency on non-dimensional vertical displacement, excess pore water pressure, vertical stress, and temperature. 

Last time, we found that normal mode analysis can eliminate the limitation which obtain the discrete error and truncation error in the numerical inverse transformation. The integral transformation method pays more attention to the evolution process of physical variables with time while the regular modal law pays more attention to the propagation characteristics of waves. We have rewritten and added some sentence to explain normal mode analysis in Part 4. However, the normal mode analysis method is mainly used for solving, and it is not obvious in the calculation result figures.

To improve the readability of the article, the discussion section have added to provide more nuanced interpretations of the results. The added parts are highlighted in blue on page 15 (lines 12-21), page 20 (lines 12-15), read as follows:

“In this paper, and are set to 1. That is, the model is degraded to the dynamic response problem of isotropic media under the integer order generalized thermoelastic theory. In addition, the problem is solved using the boundary conditions presented in [30], which results are compared with the obtained dimensionless displacement, excess pore water pressure, vertical stress, and temperature under thermal shock. It can be seen from Figs. 2-4 that, for and , the results of the two studies are mainly consistent, and the different numerical calculations cause small errors, with maximum and average deviations of 5.07% and 2.5%, respectively. The correctness and rationality of the calculated results are further verified.

It can be observed from Fig. 15 that the increase of significantly increases the excess pore water pressure, which is mainly due to the fact that the pore water has not been discharged in time after the increase of .”

Reference

[A-1]Alharbi AM, Othman MI A, Abd-Elaziz EM. 2-D Analysis of generalized thermoelastic porous medium under the effect of laser pulse and microtemperature, International Journal of Structural Stability and Dynamics, 2021, 21: 2150126.

[A-2]Bayones FS, Abd-All AM, et al. Effect of a magnetic field and initial stress on the P-waves in a photothermal semiconducting medium with an internal heat source, Mechanics Based Design of Structures and Machines, 2021, DOI: 10.1080/15397734.2021.1872384.

[A-3] Mondal S, Othman MIA. Memory dependent derivative effect on generalized piezo-thermoelastic medium under three theories, Waves in Random and Complex Media, 2020, DOI: 10.1080/17455030.2020.1730480.

[A-4]Abdou MA, Othman MIA, et al. Exact solutions of generalized thermoelastic medium with double porosity under L-S theory, Indian Journal of Physics, 2020, 94(5): 725-736.

[A-5] Kadian A, Kumar S, Kalkal KK. Magneto-thermoelastic interactions in a rotating functionally graded half-space with microtemperatures, Waves in Random and Complex Media, 2021, DOI: 10.1080/17455030.2020.1870760.

[A-6]Abo-Dahab SM, Kumar A, Ailawalia P. Mechanical changes due to pulse heating in a microstretch thermoelastic half-space with two-temperatures, Journal of Applied Science and Engineering, 2020, 23: 153161.

[A-7] Alharbi AM. The effect of diffusion on micropolar thermoelastic medium under 3PHL model, ZAMM-Zeitschrift Fur Angewandte Mathematik Und Mechanik, 2021, DOI10.1002/zamm.202100004.

[A-8]Alharbi AM, Said SM, Othman MIA. The effect of multi-phase-lag and Coriolis acceleration on a fiber-reinforced isotropic thermoelastic medium, Steel and Composite Strctures, 2021, 39: 125-134. 

5. **Conclusion:**

Summarize the major findings and the broader implications of the enhanced study. Highlight the significance of the refined numerical approach.

Response: 

The Conclusion part have been rewritten. The rewritten parts are highlighted in blue on page 22 (lines 20-29), page 23 (lines 1-9), read as follows:

(1)“The normal mode analysis method is successfully applied to the solution of the eighth-order characteristic equation of anisotropic media. It can accurately analyze all the considered distributions of physical variables.

(2)The frequency plays a crucial role for all the dimensionless physical variables regardless of the load type. In particular, it affects the distribution of various physical variables when considering the influence of thermal loads.

(3)The fractional order coefficient has only a clear effect on the dimensionless temperature when the mechanical load is considered on the upper surface of the subgrade. However, it have obvious effect on the dimensionless excess pore water pressure, vertical stress, and temperature when the upper surface of the subgrade is subjected to thermal load. The variation of the load frequency will affect the fractional coefficient on physical variables.

(4)The variation of the anisotropy of thermal conduction coefficient significantly affects all the considered dimensionless variables. When the anisotropy of thermal conduction coefficient increases, all the physical variables gradually increase. The existence of fractional order parameters makes this effect more obvious, especially in the dimensionless temperature curve under mechanical load, the dimensionless excess pore water pressure curve, and the vertical stress curve under thermal source.

”

6. **Language and Clarity:**

Review the paper for language clarity, grammar, and overall readability. Ensure that the flow of the paper is logical and coherent, with smooth transitions between sections.

Response: 

 The English style and mathematical formulation has been carefully and extensively modified.

---

## [Decision Letter · Decision Letter 1]

10 Jan 2024

Coupling Dynamic Response of Saturated Soil with Anisotropic Thermal Conductivity under Fractional Order Thermoelastic Theory

PONE-D-23-25844R1

Dear Dr. Yu,

We’re pleased to inform you that your manuscript has been judged scientifically suitable for publication and will be formally accepted for publication once it meets all outstanding technical requirements.

Kind regards,

Ghulam Rasool

Academic Editor

PLOS ONE

Additional Editor Comments (optional):

I accept the paper in its present form.

Reviewers' comments:

Reviewer's Responses to Questions

**Comments to the Author**

1. If the authors have adequately addressed your comments raised in a previous round of review and you feel that this manuscript is now acceptable for publication, you may indicate that here to bypass the “Comments to the Author” section, enter your conflict of interest statement in the “Confidential to Editor” section, and submit your "Accept" recommendation.

Reviewer #1: All comments have been addressed

Reviewer #2: All comments have been addressed

2. Is the manuscript technically sound, and do the data support the conclusions?

Reviewer #1: Partly

Reviewer #2: Yes

3. Has the statistical analysis been performed appropriately and rigorously? 

Reviewer #1: Yes

Reviewer #2: N/A

4. Have the authors made all data underlying the findings in their manuscript fully available?

Reviewer #1: Yes

Reviewer #2: Yes

5. Is the manuscript presented in an intelligible fashion and written in standard English?

Reviewer #1: Yes

Reviewer #2: Yes

6. Review Comments to the Author

Reviewer #1: (No Response)

Reviewer #2: (No Response)

7. PLOS authors have the option to publish the peer review history of their article (what does this mean?). If published, this will include your full peer review and any attached files.

Reviewer #1: No

Reviewer #2: No

---

## [Editor Report · Acceptance letter]

25 Mar 2024

PONE-D-23-25844R1 

PLOS ONE

Dear Dr. Yu, 

I'm pleased to inform you that your manuscript has been deemed suitable for publication in PLOS ONE. Congratulations! Your manuscript is now being handed over to our production team.

Kind regards, 

on behalf of

Dr. Ghulam Rasool 

Academic Editor

PLOS ONE